# Comprehensive mutational analysis of the checkpoint signaling function of Rpa1/Ssb1 in fission yeast

Yong-jie Xu[1]*, Sankhadip Bhadra[1], Alaa Taha A. Mahdi[1¤], Kamal Dev[1], Ilknur Yurtsever[1], Toru M. Nakamura[2]

**1** Department of Pharmacology and Toxicology, Boonshoft School of Medicine, Wright State University, Dayton, Ohio, United States of America, **2** Department of Biochemistry and Molecular Genetics, University of Illinois at Chicago, Chicago, Illinois, United States of America

¤ Current address: Saudi Authority for Intellectual Property, 6531, Al Ulaya, Riyadh, 13321, Saudi Arabia
* yong-jie.xu@wright.edu

## Abstract

Replication protein A (RPA) is a heterotrimeric complex and the major single-strand DNA (ssDNA) binding protein in eukaryotes. It plays important roles in DNA replication, repair, recombination, telomere maintenance, and checkpoint signaling. Because RPA is essential for cell survival, understanding its checkpoint signaling function in cells has been challenging. Several RPA mutants have been reported previously in fission yeast. None of them, however, has a defined checkpoint defect. A separation-of-function mutant of RPA, if identified, would provide significant insights into the checkpoint initiation mechanisms. We have explored this possibility and carried out an extensive genetic screen for Rpa1/Ssb1, the large subunit of RPA in fission yeast, looking for mutants with defects in checkpoint signaling. This screen has identified twenty-five primary mutants that are sensitive to genotoxins. Among these mutants, two have been confirmed partially defective in checkpoint signaling primarily at the replication fork, not the DNA damage site. The remaining mutants are likely defective in other functions such as DNA repair or telomere maintenance. Our screened mutants, therefore, provide a valuable tool for future dissection of the multiple functions of RPA in fission yeast.

## Author summary

Originally discovered as a protein required for replication of simian virus SV40 DNA, replication protein A is now known to function in DNA replication, repair, recombination, telomere maintenance, and checkpoint signaling in all eukaryotes. The protein is a complex of three subunits and the two larger ones are essential for cell growth. This essential function however complicates the studies in living cells, and for this reason, its checkpoint function remains to be fully understood. We have carried out an genetic screen of the largest subunit of this protein in fission yeast, aiming to find a non-lethal mutant that lacks the checkpoint function. This extensive screen has uncovered two mutants with a partial

**Data Availability Statement:** All relevant data are within the manuscript and its supporting information files.

**Funding:** YJX received funding under grant number NIH R35GM144307 from the National Institute of General Medical Sciences. TMN received funding under grant number NIH R01GM143316 from the National Institute of General Medical Sciences. The funder had no role in study design, data collection and analysis, decision to publish, or preparation of the manuscript.

**Competing interests:** The authors have declared that no competing interests exist.

defect in checkpoint signaling when DNA replication is arrested. Surprisingly, although the two mutants also have a defect in DNA repair, their checkpoint signaling remains largely functional in the presence of DNA damage. We have also uncovered twenty-three mutants with defects in DNA repair or telomere maintenance, but not checkpoint signaling. Therefore, the non-lethal mutants uncovered by this study provide a valuable tool for dissecting the multiple functions of this biologically important protein in fission yeast.

## Introduction

The integrity of the genome is crucial for the survival of all living organisms. To maintain genome integrity, several mechanisms have evolved in eukaryotes: the accurate DNA replication machinery, repair pathways that deal with replication errors and various types of DNA damage, and the mechanisms that control telomere homeostasis. Overseeing these cellular processes is the checkpoint system that coordinates their activities with the cell cycle progression [see review [1]]. All these DNA metabolic processes involve ssDNA as a transient intermediate that has to be recognized and properly protected from nuclease attack. Due to its abundance and high affinity for ssDNA, RPA is the major ssDNA binding protein in eukaryotes [see reviews [2,3]]. RPA is also known as replication factor A (RFA) and ssDNA binding protein (SSB). It was originally purified as a protein required for replication of simian virus SV40 DNA *in vitro*. It is now known that in addition to DNA replication, RPA is required for DNA repair, recombination, telomere maintenance, and checkpoint signaling. All these functions depend on the abilities of RPA to interact dynamically with ssDNA as well as other proteins.

RPA is a stable complex of three subunits Rpa1 (~70kDa), Rpa2 (~32kDa), and Rpa3 (~14kDa) that are all highly conserved in eukaryotes. Structural and biochemical studies have revealed six OB-fold domains designated as DNA binding domains (DBDs) in RPA. While the large subunit Rpa1 has four DBDs (A-C and F), Rpa2 and Rpa3 have only one DBD, DBD-D and -E, respectively, in each subunit. In addition to the DBD-D, Rpa2 has an N-terminal phosphorylation domain and a C-terminal winged helix domain for protein-protein interactions. Stable binding of RPA to ssDNA likely involves only three to four DBDs (A-D) that have higher affinities with DNA [see reviews [2,3]].

RPA has multiple checkpoint functions through interacting with various DNA damage response proteins, particularly those that interact with the N-terminal DBD-F in Rpa1 [see review [4]]. In mammalian cells, this domain of Rpa1 is known to interact with p53 [5], Mre11-Rad50-Nbs1 complex [6], and ATRIP [7]. ATRIP is the binding partner of ATR checkpoint sensor kinase [8]. Deletion analysis has revealed a conserved RPA binding domain in the N-terminus of ATRIP [9]. The RPA-coated ssDNA formed at perturbed replication fork or DNA damage site is believed to serve as a platform that recruits ATR-ATRIP and other damage response proteins [5,7]. The recruited ATR-ATRIP initiates the checkpoint signaling in both the DNA replication checkpoint (DRC) pathway at the forks and the DNA damage checkpoint (DDC) pathway at the DNA damage sites [10]. The RPA-ssDNA platform also promotes the loading of the Rad9-Rad1-Hus1 (911) checkpoint clamp at the 5' end of the ssDNA/dsDNA junction [11,12]. The loaded 911, once phosphorylated by ATR, recruits more proteins such as the checkpoint adaptor protein TopBP1. The recruited TopBP1 can stimulate ATR kinase via its ATR activation domain and thus enhance the checkpoint signaling [13]. Like TopBP1, ETAA1 also activates mammalian ATR kinase both *in vitro* and *in vivo* [14] similar to the budding yeast Ddc2, Dna2, and Dpb11 that activate Mec1 kinase, the ATR ortholog

[15,16,17]. Proteomics analyses have identified hundreds of phosphorylation targets of ATR in both mammalian and yeast cells [18,19,20], including RPA [21,22]. Phosphorylation of Rpa2 by ATR at the phosphorylation domain is believed to regulate the functions of RPA [23].

Most of the yeast genetic studies of RPA are carried out with the budding yeast *S. cerevisiae*. Early studies showed that all three RPA subunits are essential for cell survival and the two larger subunits are the phosphorylation targets of Mec1 in budding yeast [21,22,24,25]. Although deletion of *ssb3* gene encoding the small subunit in fission yeast is not lethal, the null mutant is sensitive to the genotoxins that disrupt DNA replication [26]. Due to the essentiality of RPA, studying its functions *in vivo* has been challenging, and relying on the mutants that allow cell survival and, in the meantime, are defective in checkpoint, or other functions. One such separation-of-function mutant is the budding yeast *rfa1-t11*, which was first reported 25 years ago [27]. This mutant carries a single mutation that converts Lys$^{45}$ residue to glutamic acid in the N-terminal DBD-F domain. The *rfa1-t11* mutant is defective in homologous recombination [28,29] and Mec1-mediated checkpoint signaling [7]. Another *rfa1* allele identified by an earlier study also showed a defect in the DRC [30]. Since the *rfa1-t11* mutation is in the DBD-F, it is generally believed that the mutation interrupts the interaction between RPA and Ddc2, the ATRIP homolog in budding yeast, leading to the defect in Mec1 kinase signaling. However, not all previous studies are consistent with the checkpoint sensor function of RPA [25,31,32,33]. A more recent structural study showed that while the N-terminus of Ddc2 binds to the DBD-F of Rfa1, the *rfa1-t11* mutation, however, does not significantly affect its binding to Ddc2 [34] although the same mutation affects its binding to Mre11-Rad50-Xrs2 complex [35]. This suggests that the checkpoint functions of RPA, particularly its checkpoint sensor function in the DRC pathway, remain to be fully understood *in vivo*.

The fission yeast *S. pombe* is an established model for studying the cellular mechanisms that are conserved in higher eukaryotes. Unlike the budding yeast in which the major checkpoint effector kinase Rad53 (functional homolog of human Chk1) activates both the DRC and the DDC pathways, the DRC and the DDC pathways are mediated by Cds1 (human Chk2) and Chk1 [36] separately, in fission yeast, which promotes unambiguous description of the checkpoint signaling mechanisms. Several RPA mutants have been reported previously in fission yeast that are sensitive to ionizing radiation and genotoxic drugs [26,37,38]. None of them however appears to have a defined checkpoint defect. We have recently carried out a large-scale genetic screen in fission yeast by random mutation of the genome, looking for mutants that are defective in the DRC signaling pathway. This hydroxyurea (HU)-sensitive or *hus* screen has identified several previously uncharacterized mutants such as *tel2-C307Y* in the essential Tel2-Tti1-Tti2 complex [39], a series of mutants of *rqh1* of a RecQ helicase [40], and two mutants of the essential Smc5/6 complex [41]. To our surprise, although this genome-wide screen has identified every single previously known DRC gene multiple times, it did not identify a single RPA mutant, which raises a concern about the checkpoint sensor function of RPA *in vivo*. To better understand the *in vivo* checkpoint functions of RPA, we took a targeted forward genetics approach to screen mutants in the large subunit Ssb1, aiming to identify a non-lethal mutant that lacks the checkpoint signaling function. Such a mutant, once identified, would provide a much clearer insight into the checkpoint initiation mechanisms at the replication fork. After an extensive screen, we report here our identification of two mutants *ssb1-1* and *ssb1-10* that are sensitive to HU and various other genotoxins. The two mutants are confirmed partially defective in checkpoint signaling primarily in the DRC, not the DDC pathway. We have also screened twenty-three other *ssb1* mutants whose defects are likely in DNA repair or telomere maintenance. Together, these identified *ssb1* mutants provide a valuable tool for future studies of the multiple functions of RPA in fission yeast.

## Results

### Minimal checkpoint defects in the previously reported *S. pombe* RPA mutants

Rad3 is the ortholog of human ATR and the *S. cerevisiae* Mec1 in fission yeast. It is the master checkpoint sensor kinase that activates both the DRC and the DDC pathways. Tel1, the ATM homolog, contributes minimally to the checkpoint functions in *S. pombe*. In the DRC pathway, Rad3 activates the effector kinase Cds1 (hCHK2/scRad53) at the perturbed replication fork. When DNA is damaged, Rad3 activates Chk1 in the DDC pathway to induce the DNA damage response, which mainly occurs at G2 as it is a major phase (~75%) of the cell cycle in fission yeast. Mutants of the DRC and DDC pathways are highly sensitive to replication stress and DNA damage. Earlier studies have identified several RPA mutants in *S. pombe* that are sensitive to DNA damage [26,37,38,42]. However, none of the mutants appears to have a checkpoint defect. We obtained four RPA mutants *rad11A*, *ssb1-418(G78E)*, *ssb1-D223Y*, and *ssb3Δ*, and examined their Rad3-mediated checkpoint signaling. Since the mutation in *rad11A* was unknown, we sequenced the *ssb1* genomic locus and identified a single missense mutation that changes Arg[339] to histidine. The *rad11A* mutant is hereafter renamed as *ssb1-R339H* in this study.

We first examined the sensitivities of the RPA mutants to HU and the DNA-damaging agent methyl methanesulfonate (MMS) by spot assay. HU depletes cellular dNTPs and generates replication stress by slowing the polymerase movement at the fork. Since the DRC deals with the replication stress, cells lacking Cds1 are highly sensitive to HU (Fig 1A). In the presence of MMS, cells lacking Chk1 are highly sensitive, indicating that the DNA damage is mainly dealt with by the DDC. Since the *rad3Δ* mutant lacks both the DRC and the DDC, it is highly sensitive to both HU and MMS (Fig 1A). Under similar conditions, the four RPA mutants were found sensitive to both HU and MMS, particularly the *ssb1-R339H* mutant.

We then examined the Rad3-initiated checkpoint signaling in these RPA mutants by Western blotting using the phospho-specific antibodies described in our previous studies [43,44]. In the presence of replication stress, Rad3 phosphorylates Thr[645] and Thr[653] in the middle of Mrc1 [45], the mediator of the DRC. The two phosphorylated residues function redundantly to recruit Cds1 to be phosphorylated by Rad3 [46]. Phosphorylation of Cds1-Thr[11] by Rad3 promotes homodimerization of inactive Cds1, which stimulates Cds1 autophosphorylation at Thr[328] in the activation loop [44]. Autophosphorylation of Thr[328] directly activates Cds1 and the activated Cds1 mediates most of the biological functions of the DRC in fission yeast [44]. As shown in Fig 1B, when wild-type cells were treated with (+) or without (-) HU, Rad3-dependent phosphorylation of Mrc1 was significantly increased. Since Mrc1 is expressed during the G1/S phase and the activated DRC promotes Mrc1 expression [45,47], the protein level of Mrc1 is higher in HU-treated wild-type cells than in untreated cells and less increased in HU-treated *rad3Δ* cells. Under similar conditions, Mrc1 phosphorylation in the RPA mutants was examined and compared with that in wild-type cells. The experiment was repeated three times and the quantitation results are shown in Fig 1C. We found that in the presence of HU, Rad3 phosphorylation of Mrc1 was unaffected in *ssb1-R339H*, *ssb1-D223Y*, *ssb3Δ* but moderately reduced in *ssb1-G78E* mutant. When Rad3 phosphorylation of Cds1 was examined in the presence of HU, we found that Cds1 phosphorylation was unchanged in *ssb1-G78E*, moderately reduced in *ssb3Δ*, or even higher in *ssb1-R339H* and *ssb1-G78E* (Fig 1D and 1E). These results show that although the mutants are sensitive to HU, these mutants do not significantly compromise the Rad3 kinase signaling in the DRC pathway.

Since the RPA mutants are sensitive to MMS (Fig 1A), we next examined the phosphorylation of Chk1 by Rad3 [48,49], which activates Chk1 in the DDC pathway. Chk1

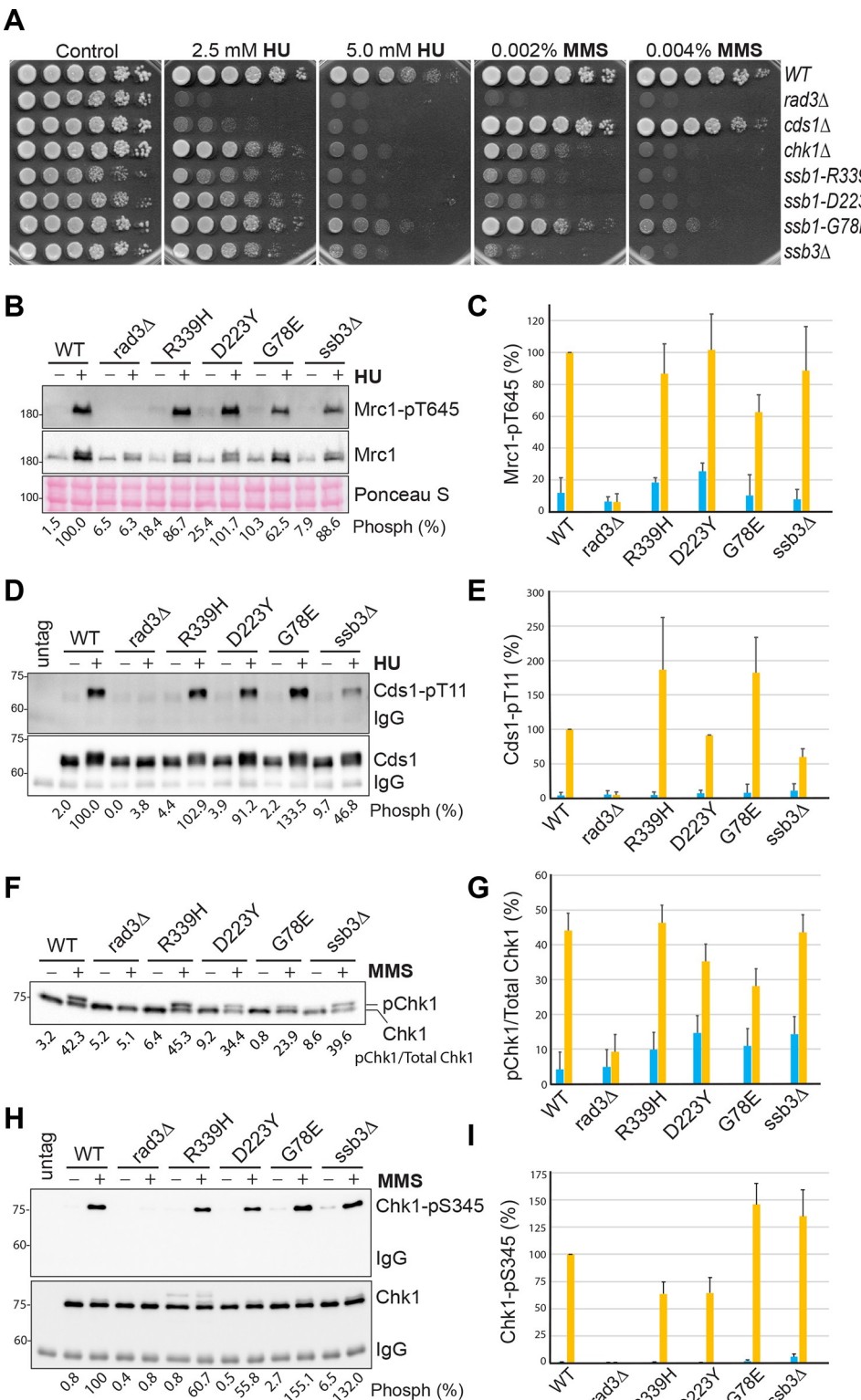

**Fig 1. Functional checkpoint signaling in the previously reported RPA mutants in *S. pombe*.** (**A**) Sensitivities of the four previously reported RPA mutants to HU and MMS were examined by spot assay. A series of five-fold dilutions of the logarithmically growing cells were spotted on YE6S plates or plates containing HU or MMS. The plates were incubated at 30°C for 3 days and then photographed. Wild-type (TK48) cells and the checkpoint mutants *rad3Δ* (NR1826), *cds1Δ* (GBY191), and *chk1Δ* (TK197) were included as controls. (**B**) Phosphorylation of Mrc1 by Rad3 was

unaffected or moderately reduced in the four RPA mutants. Wild type and the mutant cells used in A were treated with (+) or without (-) 15 mM HU for 3 h. Phosphorylation of Mrc1 (upper panel) was examined by Western blotting of whole cell lysates made from the TCA-fixed cells after SDS-PAGE using the phospho-specific antibody. The same blot was stripped and reprobed with anti-Mrc1 antibodies (middle panel). A section of the Ponceau S-stained membrane is shown for loading control (bottom panel). The phosphorylation bands were quantified, and the intensities relative to the HU-treated wild-type cells are shown at the bottom. (**C**) The Western blotting shown in B was repeated three times and the quantitation results are shown in percentages. Error bars represent the means and SDs of the triplicates. Blue and brown columns indicate before and after HU treatment, respectively. (**D**) Phosphorylation of Cds1 by Rad3 was increased or moderately reduced in the four RPA mutants. Wild type and the indicated mutant cells were treated with HU as in B. Cds1 was IPed and analyzed by Western blotting using an anti-HA antibody (bottom panel). The same membrane was stripped and then blotted with the phospho-specific antibody (upper panel). The phosphorylation bands were quantified and relative intensities are shown at the bottom. (**E**) The experiments in D were repeated three times and the quantitation results are shown. (**F**) Chk1 phosphorylation was examined in wild-type and the mutant cells treated with (+) or without (-) 0.01% MMS for 90 min. The whole cell lysates made by the TCA method were analyzed by SDS-PAGE followed by Western blotting with anti-HA antibody. (**G**) Quantitation results from three separate blots as in F are shown in ratios of phosphorylated Chk1 vs total Chk1. (**H**) Chk1 phosphorylation was examined by Western blotting using the phospho-specific antibody. Wild type and the indicated mutant cells were treated MMS as in F. Chk1 was IPed and then analyzed by Western blotting using the antibody against Chk1-pS345 (top panel). The same membrane was stripped and blotted with an anti-HA antibody (bottom panel). The relative intensities of the Chk1-pS345 bands were quantified, normalized with that of Chk1 bands, and shown in percentages. (**I**) Quantitation results from three repeats of H are shown.

phosphorylation is commonly examined by mobility shift assay [48]. Using this assay, we found that after treatment with MMS, Chk1 phosphorylation was significantly increased in wild-type cells and the phosphorylation was dependent on Rad3 (Fig 1F). When Chk1 phosphorylation was examined in the four RPA mutants, we found it was unaffected in *ssb1-R339H* and *ssb3Δ* or slightly to moderately reduced in *ssb1-D223Y* and *ssb1-G78E*, respectively (Fig 1G). In addition to the phosphorylation of Chk1-Ser$^{345}$, which is crucial for Chk1 activation, Rad3 also phosphorylates other residues on Chk1 such as Ser$^{323}$ and Ser$^{367}$ [48,49] that may affect the mobility shift of phosphorylated Chk1. To preclude this possibility, we generated a phospho-specific antibody for Chk1-pS345. The specificity of the antibody was confirmed by Western blotting of the immunopurified (IPed) Chk1 from the MMS-treated cells (S1 Fig). Using this antibody, we examined Chk1 phosphorylation in the four RPA mutants (Fig 1H and 1I). The results showed that when treated with MMS, Chk1 phosphorylation was moderately reduced in *ssb1-R339H* and *ssb1-D223Y* or even increased in *ssb1-G78E* and *ssb3Δ* mutants. The differences between the results by the mobility shift assay (Fig 1F and 1G) and by the phospho-specific antibody (Fig 1H and 1I) are likely due to the different antibodies and quantitation methods used and the potential issues with the loading, particularly the IPed Chk1 (see Discussion). We conclude that the Rad3 kinase signaling in the DRC and the DDC pathways are minimally compromised or remain functional in the four previously reported RPA mutants.

## Insensitivity of *ssb1-R339H*, *ssb1-D223Y*, *ssb1-G78E*, and *ssb3Δ* to acute HU treatment

Since the DRC remains functional in the RPA mutants, we wanted to investigate their HU sensitivities. In addition to the replication stress, HU induces other types of cellular stress such as oxidative stress, particularly under chronic exposure conditions such as the spot assay shown in Fig 1A. Some metabolic mutants in *S. pombe* are highly sensitive to chronic HU exposure but resistant to MMS and the acute treatment with HU [50,51]. Since mutations in replication genes can increase oxidative stress [52] and thus sensitizes the cells to chronic HU exposure, we then examined the sensitivity of the four RPA mutants to acute HU treatment in liquid cultures, which mainly generates the replication stress, not the oxidative stress. When cell recovery from the acute HU treatment was examined by spot assay (Fig 2A), we found that unlike

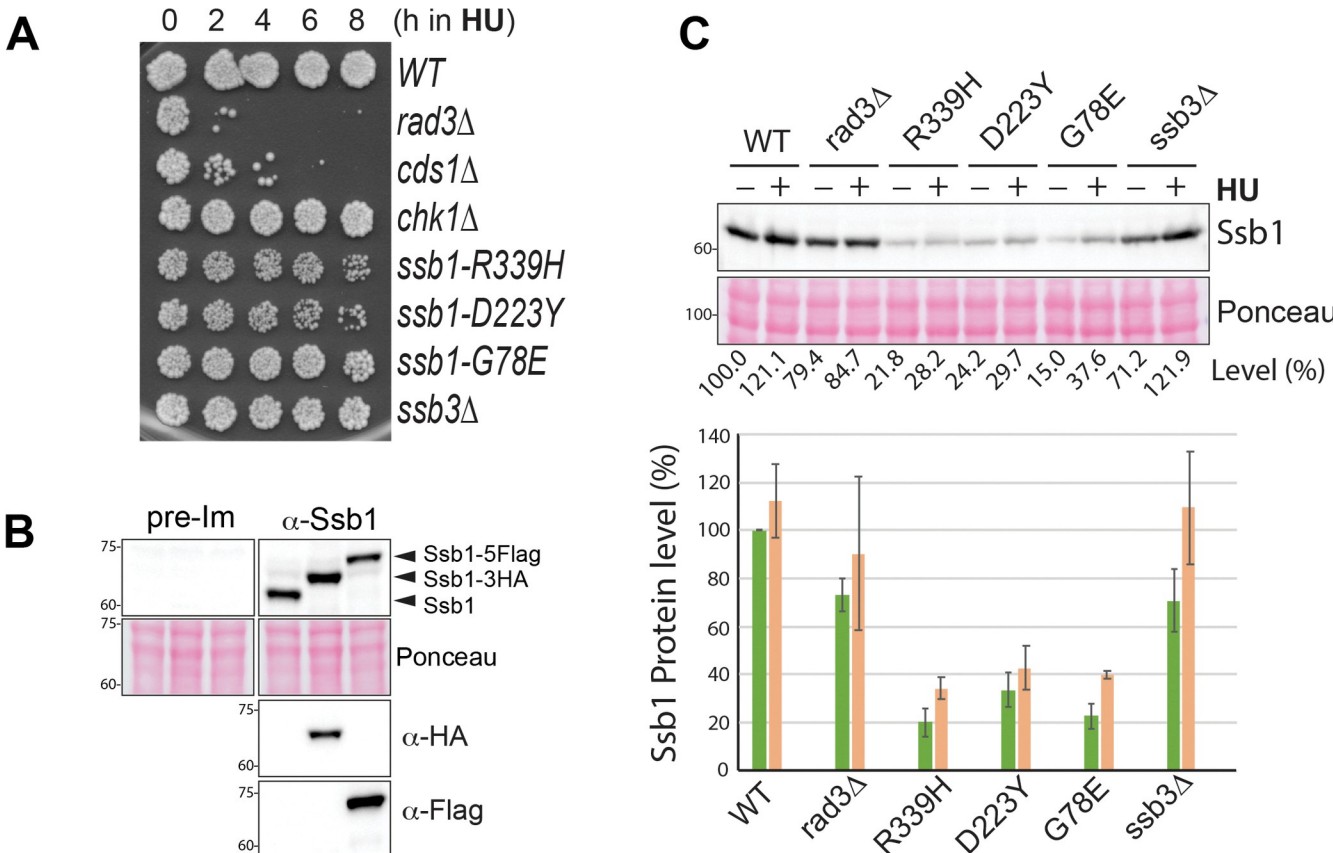

**Fig 2. Insensitivity to acute HU treatment and the Ssb1 levels in the previously reported *S. pombe* RPA mutants. (A)** Sensitivities of the RPA mutants to acute HU treatment were determined by spot assay as described in Materials and Methods. (**B**) Specificity of the antibody against Ssb1. *S. pombe* expressing with untagged or tagged Ssb1 were lysed and analyzed by Western blotting using the anti-Ssb1 antibody or pre-immune antiserum (upper panels). The Ponceau S-stained membrane is shown for loading (middle panels). The same blot was stripped and reblotted sequentially with anti-HA and then anti-Flag antibodies to confirm the tagged Ssb1 (lower two panels). (**C**) Ssb1 levels were examined by Western blotting in wild type and the indicated mutant cells before (-) or after (+) HU treatment (upper panels). Loading is shown by Ponceau S staining. The relative levels of Ssb1 are shown at the bottom. The lower panel is the quantitation results of three separate blots.

the *rad3Δ* and *cds1Δ* mutants of the DRC pathway that died within 4 h of the HU treatment, the *chk1Δ* mutant in the DDC pathway and the four RPA mutants were relatively insensitive, although the *ssb1-R339H* and *ssb1-D223Y* mutants showed a growth defect and a minimal HU sensitivity, which is consistent with their defects in DNA repair. Thus, the chronic HU sensitivity observed in the four RPA mutants in Fig 1A is likely a combinatory effect of the minimal replication defects caused by the mutations, potential defects in DNA repair, and the oxidative stress induced by the chronic exposure to HU. The relative insensitivity of these RPA mutants to acute HU treatment provides additional support to the conclusion that the mutations do not significantly compromise the DRC.

## Destabilized Ssb1 in the *R339H*, *D223Y*, and *G78E* mutants

RPA is essential for cell growth and perturbation of its protein level causes genome instability or even cell death [53]. To better understand the drug sensitivities of the RPA mutants, we generated an antibody against Ssb1. The specificity of the antibody was confirmed by Western blotting of the *S. pombe* whole cell lysates after SDS-PAGE. As shown in Fig 2B, the antibody, but not the pre-immune serum, detected a strong band of the expected size for Ssb1. The

specificity of the antibody was further confirmed by detecting the upper-shifted Ssb1 with a C-terminal 3HA or a 5Flag tag. Using this specific antibody, we examined the Ssb1 levels in *S. pombe* treated with (+) or without (-) HU (Fig 2C). HU treatment slightly increased Ssb1 levels in wild-type, *rad3Δ*, as well as the four RPA mutants, which is consistent with its main function during DNA replication. Compared with wild-type cells, the Ssb1 level is moderately reduced in *rad3Δ* cells, suggesting that the checkpoint may contribute to RPA homeostasis under normal or stress conditions. The Ssb1 level in the *ssb3Δ* cells is moderately reduced under normal conditions and higher in HU. On the other hand, the Ssb1 levels in *ssb1-R339H*, *ssb1-D223Y*, and *ssb1-G78E* mutants were significantly reduced to $\leq 40\%$ of the wild-type level. Thus the reduced Ssb1 level could be a contributing factor for the sensitivities of the three RPA mutants to HU and MMS shown in Fig 1A. Altogether, our results show that the Rad3 kinase signaling remains functional in the four previously reported RPA mutants, which confirms the previous cell biological studies for *ssb1-R339H* (*rad11A*) and *ssb3Δ* mutants [26,37].

## Integration of the *rfa1-t11* mutation in *S. pombe* affects cell survival

Earlier studies in budding yeast showed that the *rfa1-t11(K45E)* mutation compromises the Mec1 kinase signaling [7,54]. Since the mutated residue Lys[45] in the N-terminal DBD-F of the budding yeast Rfa1 is highly conserved in eukaryotes (Fig 2A), we decided to make three similar charge reversal mutations *K45E*, *R46E*, and *K45E-R46E* in *S. pombe ssb1*, integrate them at the genomic locus and then examine whether any of the mutations compromise the Rad3 kinase signaling in *S. pombe*. To facilitate the integration, we made a strain in which *ssb1* is tagged with a C-terminal HA linked to *ura4* marker as diagrammed in S2B Fig. The C-terminal tagging and the linked *ura4* were confirmed by colony PCR, tetrad dissection after back-crossing with the parental strain (S2C Fig), and Western blotting of the whole cell lysates with an anti-HA antibody (S2D Fig). Using this strain, we integrated the three mutations at the genomic locus by the marker switching method (S2E Fig). We found that although integration of *K45E* and *K45E-R46E* mutations did not yield any colonies, integration of the *R46E* mutation formed colonies, but the colony sizes were much smaller. This result suggests that while the mutation of Lys[45] is lethal, the mutation of the conserved Arg[46] significantly compromises cell growth in fission yeast. We confirmed this result by spot assay (S2F Fig). Due to the severe growth defect, we did not pursue the *ssb1-R46E* mutant.

## Screening of non-lethal *ssb1* mutants that are sensitive to HU and MMS

The results described above clearly showed that a separation-of-function mutant of RPA that lacks the checkpoint signaling function remains unavailable in *S. pombe*. Since Ssb3 does not contribute to checkpoint signaling and previous studies in *S. cerevisiae* suggest that the large subunit of RPA plays an important role in checkpoint signaling, we decided to use the targeted forward genetics approach [55] to screen new *ssb1* mutants lacking the checkpoint signaling function. Such a mutant, once uncovered, would provide a much clearer picture of checkpoint initiation mechanisms in cells. For the screening, we cloned the *ssb1* expression cassette into the pJK210 integration vector [56] that carries the *ura4* marker (S3A Fig). For the convenience of random mutation, a silent mutation was introduced in the middle of *ssb1* to generate a SpeI restriction site (Fig 3A). Random mutations were then generated by PCR [57] in the N- and the C-terminal halves of *ssb1* in two separate libraries. Allele replacement at the *ssb1* genomic locus was achieved by transforming wild-type *S. pombe* lacking the *ura4* gene with linearized library DNA. The transformed cells with integrated *ura4* gene were selected during the pop-in step by sequential culturing in EMM6S[ura-] medium (S3A Fig, Step 1 and 2). To allow the

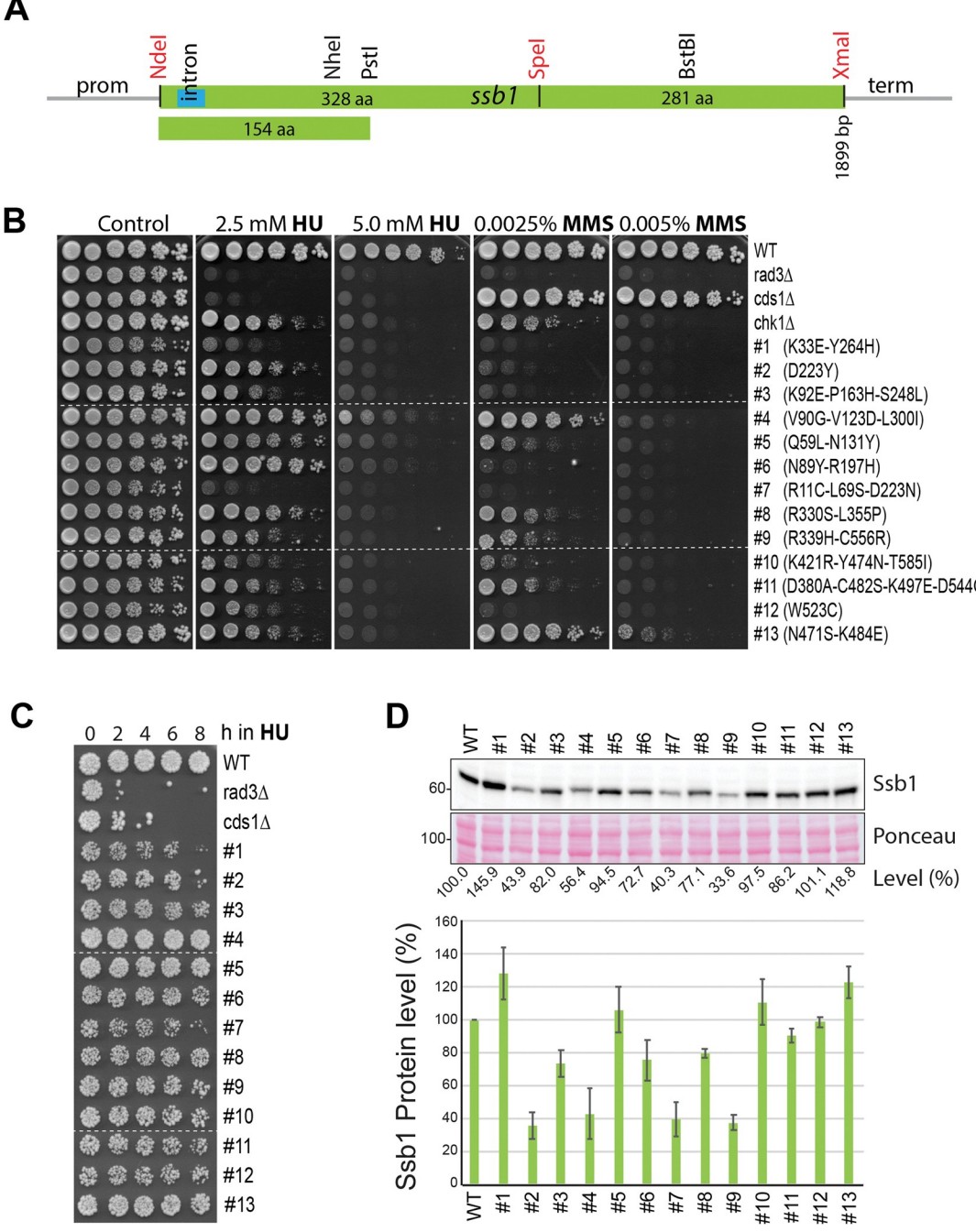

**Fig 3. Targeted genetic screen identified twenty-five *ssb1* primary mutants. (A)** Diagram of the *ssb1* expression cassette. A small intron near the 5' end is shown in blue. The restriction sites NdeI, SpeI, and XmaI made by point mutations are marked in red whereas the endogenous sites are indicated in black. Random mutations were made by PCR between NdeI and SpeI, SpeI and XmaI, and NdeI and PstI sites in three separate libraries. Precise replacement of the mutated alleles at the genomic locus was carried out in two steps as described in the Materials and Methods. **(B)** Sensitivities of the 1st set of thirteen primary *ssb1* mutants to HU and MMS were determined by spot assay as in [Fig 1A](). The amino acid changes caused by the mutations are shown on the right. Dashed lines indicate discontinuity of original plates used to generate the figure. Note: the 2nd set of twelve *ssb1* primary mutants identified by screening the N-terminal 154 amino acid region is shown in [S4]() and [S5]() Figs. **(C)** Sensitivities of the thirteen mutants to acute HU treatment were determined by spot assay. Dashed lines indicate the discontinuity. **(D)** Ssb1 levels in the thirteen mutants were examined by Western blotting (upper panels). Quantitation results from three separate blots are shown in the lower panel.

loss of the integrated *ura4* gene during the second pop-out step, the cells were cultured in YE6S to saturation and then spread on 5-fluoroorotic acid (5-FOA) plates to counter-select those that had lost the *ura4* gene (S3A Fig, Step 3). Colonies formed on the 5-FOA plates were replicated onto YE6S plates containing HU. The sensitive colonies were streaked out into single colonies, tested again for their sensitivities to HU and MMS by replica plating, and the drug sensitivities were then confirmed by spot assay (S3B Fig, red asterisks). The screened mutants were backcrossed at least once before DNA sequencing to identify the mutations in *ssb1*. After redundant mutants were removed, the mutants shown in S3B Fig were renumbered. In total, thirteen primary *ssb1* mutants were screened by random mutations of the whole ORF of *ssb1*. The *ssb1* mutations identified in the thirteen mutants and their drug sensitivities determined by spot assay are shown in Fig 3B. Some of the screened mutants are temperature-sensitive (S3C Fig), which is consistent with the essential function of *ssb1*. Among the thirteen mutants, the #2 primary mutant (and two other independent mutants) carry the same mutation as the previously characterized *ssb1-D223Y* mutant (Fig 1) with defects in DNA repair and telomere maintenance [38], which suggests that our screen is extensive (see Discussion).

We then examined the sensitivity of the primary mutants to acute HU treatment by spot assay. As shown in Fig 3C, all mutants except the #4, #5 and #13 mutants showed a growth defect. Among the thirteen mutants, the #1, #2, and #7 mutants showed noticeable sensitivities, although the sensitivities were lower than *cds1Δ* cells. When the Ssb1 levels were examined in these mutants (Fig 3D), we found that Ssb1 in #2, #4, #7, and #9 mutants was reduced to ≤ 40% of the wild-type level and the rest of mutants showed a moderately reduced or slightly increased Ssb1. The increased Ssb1 likely compensates for the functional loss caused by the mutations.

Next, we examined the Rad3 signaling in the thirteen mutants by Western blotting. As shown in Fig 4A, in the presence of HU, Mrc1 phosphorylation was reduced to ~40–60% of the wild-type level in #1 and #7 mutants, which is consistent with their acute HU sensitivity (Fig 3C). The #2 mutant, like *ssb1-D223Y* (Fig 1B), showed an increased Mrc1 phosphorylation in HU, consistent with its defects in DNA repair and telomere maintenance [38], not the checkpoint signaling (Fig 1). In the remaining mutants, Mrc1 phosphorylation was either unaffected (#6 and #8) or moderately reduced (#3-#5 and #9-#13). Although there are noticeable differences, results from the Cds1 phosphorylation in the thirteen mutants (Fig 4B) are generally agreeable with that from Mrc1 phosphorylation (Fig 4A). When Chk1 phosphorylation was examined in the presence of MMS by the phospho-specific antibody, we found that while most mutants, except the #3 mutant, showed reduced Chk1 phosphorylation in a range of ~30–60% of wild-type level, and the phosphorylation was more significantly reduced in the #10 mutant (Fig 4C). To confirm this result, we examined the Chk1 phosphorylation by the mobility shift assay (Fig 4D). To our surprise, none of the thirteen mutants showed a significant defect, showing that the DDC remains largely functional in these mutants including the #7 mutant (see below). The differences between the results in Fig 4C and 4D are likely technical as mentioned above or due to the complications of secondary mutations (see Discussion).

## Screening the N-terminal region identified twelve more *ssb1* mutants

Because the genetic screen described above did not identify an *ssb1* mutant with a significant checkpoint defect and earlier studies in both mammalian and budding yeast cells have shown that the N-terminal DBD-F of RPA1 plays an important role in checkpoint initiation [7,12,54], we used the same strategy to screen more mutants by focusing on the N-terminal 154 amino acid region of Ssb1 between the NdeI and PstI sites that contains the DBD-F (Figs 3A and 5A).

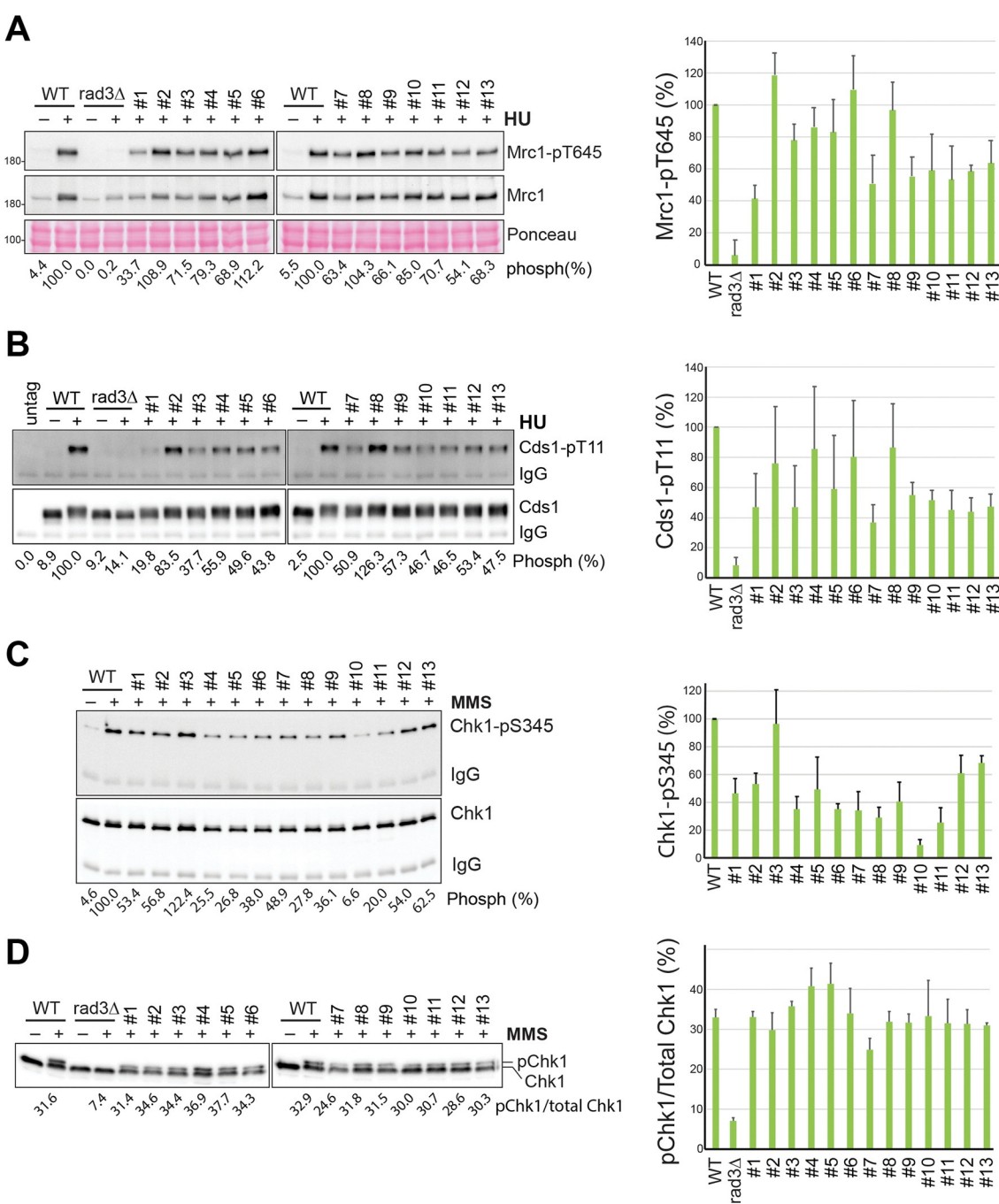

**Fig 4. Checkpoint signaling defects in the 1st set of thirteen primary *ssb1* mutants.** (**A**) Rad3-dependent Mrc1 phosphorylation in the thirteen *ssb1* mutants was examined as in Fig 1B (left panels). Quantitation results from three repeats are shown on the right. (**B**) Rad3-dependent Cds1 phosphorylation in the *ssb1* mutants was examined (left panels), repeated three times, and the quantitation results are shown on the right. (**C**) Rad3-dependent Chk1 phosphorylation in the *ssb1* mutants treated with MMS was examined by Western blotting using the phospho-specific antibody as in Fig 1H. The quantitation results are shown on the right. (**D**) The Chk1 phosphorylation in the MMS-treated *ssb1* mutants was examined by the mobility shift assay as in Fig 1F. The quantitation results are shown on the right.

This screen identified twelve more *ssb1* mutants (#14 - #25 in S4A Fig). Among these mutants, the #24 mutant showed a significant sensitivity to HU, MMS, as well as the acute treatment of HU (S4B Fig), and a moderate reduction in Ssb1 level (S4C Fig). More importantly,

**A**

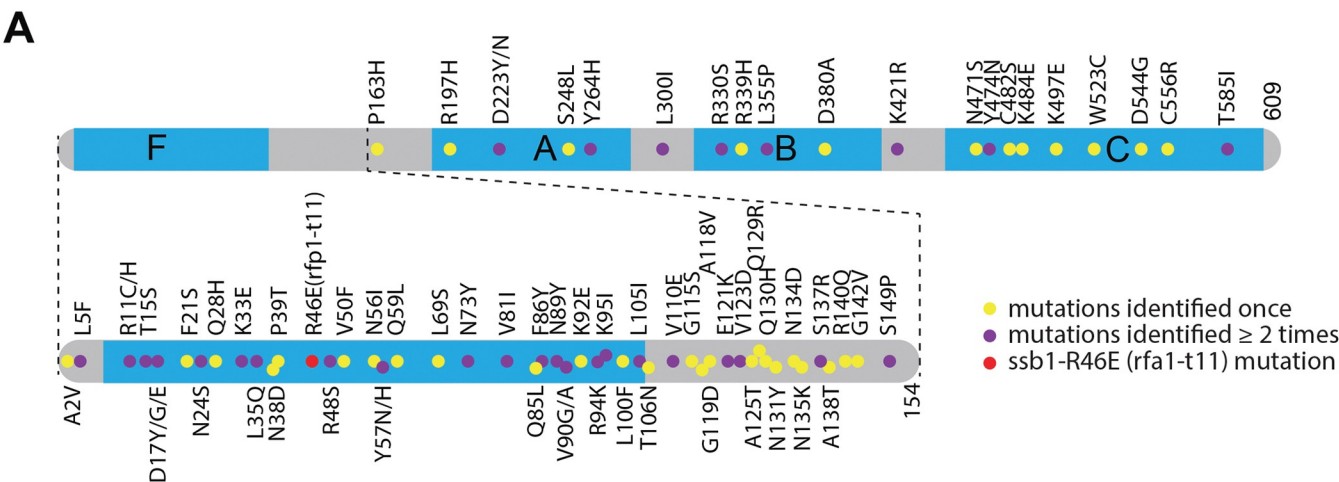

**B**

| Allele | Amino acid change | Cell growth | Chronic HU | Acute HU | MMS | Ssb1 level | Mrc1-pT645 | Cds1-pT11 | Chk1-pS345 | pChk1/Chk1 |
|---|---|---|---|---|---|---|---|---|---|---|
| WT | none | *** | R | R | R | 100 | 100 | 100 | 100 | 33.0(2.0) |
| ● #1 | K33E-Y264H (2) | * | **** | ** | **** | 128.1(15.6) | 41.5(8.3) | 47.1(22.2) | 46.3(10.8) | 33.0(1.4) |
| #2 | D223Y (3) | ** | ** | * | **** | 36.0(8.1) | 118.4(13.9) | 75.9(37.7) | 53.2(7.6) | 29.8(4.31) |
| #3 | K92E-P163H-S248L (1) | ** | *** | UD | **** | 73.3(8.1) | 77.8(10.0) | 47.0(27.4) | 96.4(24.3) | 35.7(1.2) |
| #4 | V90G-V123D-L300I (3) | *** | * | UD | * | 43.0(15.2) | 86.1(12.2) | 85.6(41.5) | 35.3(8.9) | 40.8(4.5) |
| #5 | Q59L-N131Y (1) | *** | ** | UD | *** | 106.1(13.7) | 83.0(20.2) | 58.9(35.6) | 49.4(23.1) | 41.3(5.2) |
| #6 | N89Y-R197H (1) | ** | * | UD | **** | 75.7(12.2) | 109.5(21.4) | 80.4(37.2) | 35.2(3.8) | 33.9(6.2) |
| ● #7 | R11C-L69S-D223N (1) | * | **** | ** | **** | 39.9(10.5) | 50.7(17.8) | 36.9(11.5) | 34.0(13.6) | 24.9(2.9) |
| #8 | R330S-L355P (3) | ** | * | UD | *** | 79.7(2.8) | 96.6(17.5) | 86.5(29.1) | 29.1(7.3) | 31.9(2.6) |
| #9 | R339H-C556R (1) | ** | ** | UD | *** | 37.8(4.5) | 55.3(12.3) | 54.9(8.5) | 40.6(13.7) | 31.6(2.2) |
| ● #10 | K421R-Y474N-T585I (4) | ** | **** | UD | *** | 110.6(13.8) | 59.1(22.5) | 51.8(6.2) | 9.2(4.1) | 33.3(8.9) |
| #11 | D380A-C482S-K497E-D544G (1) | ** | *** | UD | *** | 90.6(4.3) | 53.4(21.1) | 45.2(12.7) | 25.5(10.5) | 31.5(6.0) |
| #12 | W523C (1) | ** | *** | UD | **** | 98.9(3.1) | 58.6(3.9) | 44.0(9.3) | 60.8(12.8) | 31.4(3.6) |
| #13 | N471S-K484E (1) | *** | *** | UD | * | 123.1(9.6) | 63.7(14.1) | 47.2(8.5) | 68.2(5.1) | 31.0(0.6) |
| #14 | L35Q-S149P (2) | * | *** | UD | **** | 109.0(4.1) | 55.7(24.5) | 33.4(22.9) | 127.5(49.1) | 40.7(2.6) |
| #15 | L5F-N24S-R48S-F86Y-V90A (2) | * | *** | UD | **** | 76.4(4.1) | 54.1(21.0) | 46.4(14.0) | 155.3(65.6) | 38.2(0.5) |
| #16 | E121K (1) | ** | * | UD | *** | 59.6(12.7) | 72.6(16.1) | 111.8(25.9) | 109.3(62.7) | 37.9(3.2) |
| ● #17 | Y57N-Q130H-N134D (1) | ** | *** | UD | **** | 92.4(10.4) | 83.0(30.0) | 73.5(8.1) | 128.6(202.9) | 41.4(1.3) |
| #18 | R11H-T15S-D17Y-N73Y-K95I-L105I (2) | * | *** | * | **** | 47.3(8.2) | 50.0(6.4) | 46.7(25.9) | 98.9(64.5) | 32.0(2.0) |
| ● #19 | L35Q-P39T-N56I-G142V (1) | ** | *** | UD | **** | 77.2(13.6) | 68.9(29.6) | 32.8(10.3) | 254.6(194.2) | 41.4(2.4) |
| #20 | D17G-F21S-V50F-G115S-N135K-R140Q (1) | ** | * | UD | * | 73.7(16.6) | 115.8(31.2) | 48.5(3.0) | 153.3(115.9) | 47.4(2.0) |
| #21 | Y57H-V81I-R94K-V110E-S137R (3) | * | ** | * | ** | 58.1(7.9) | 94.0(54.3) | 104.4(17.6) | 103.2(68.1) | 40.8(1.9) |
| #22 | Q28H-L35Q-A125T-A138T (1) | *** | ** | UD | ** | 58.4(6.0) | 100.1(15.6) | 40.2(11.2) | 148.2(95.5) | 38.6(2.8) |
| #23 | N38D-Q85L-T106N-E121K-Q129R (1) | *** | ** | UD | *** | 40.1(5.1) | 83.3(22.8) | 118.8(30.9) | 85.9(50.3) | 37.0(1.4) |
| ● #24 | L100F-G119D (1) | * | **** | *** | **** | 66.3(4.8) | 53.7(33.2) | 19.7(7.4) | 86.9(51.1) | 41.7(4.3) |
| #25 | A2V-D17E-N89Y-A118V (1) | *** | *** | UD | **** | 52.1(10.7) | 75.2(25.8) | 152.3(97.2) | 139.3(64.5) | 38.4(4.2) |

WT: 43.1(4.7)

● with partial checkpoint signaling defects in the replication checkpoint pathway
● with secondary mutations
● with functional checkpoints

**Fig 5. Summary of the twenty-five *ssb1* primary mutants identified by this extensive screen.** (A) Diagram of Ssb1 and the relative positions of amino acid changes caused by the mutations. The four DNA binding domains F, A, B, and C are shown in blue. The intensively screened N-terminal region containing the F domain is enlarged. Dots indicate the relative locations of the mutated amino acid residues. While the yellow dots indicate the mutations that were identified once, the purple dots are those that were identified at least two times in separate mutants. The red dot indicates *ssb1-R46E* mutation that is analogous to the budding yeast *rfc1-t11* in *S. pombe*. (B) The cell growth, drug sensitivities, Ssb1 levels, and checkpoint defects of the twenty-five primary *ssb1* mutants identified in this study. The number of the primary mutants and their mutations are shown in the 1st and 2nd columns from the left, respectively. Numbers in parentheses indicate the times the mutants were independently screened. Asterisks in the 3rd column indicate the relative cell growth status estimated on YE6S plates in the spot assays (Figs 3B and S4A). Relative sensitivities to chronic (Figs 3B and S4A) and acute HU treatment (Figs 3C and S4B) determined by spot assay are shown by the asterisks in the 4th and 5th columns, respectively. R: resistance; UD: undetectable or minimal sensitivity. Relative Ssb1 levels in logarithmically growing cells were shown in the 7th column. The numbers in parentheses are SD values of three repeats. Similarly, phosphorylation Mrc1 and Cds1 in HU are shown in the 8th and 9th columns, respectively. Chk1 phosphorylations determined by phospho-specific antibody and the mobility shift assay are shown in the 10th and 11th columns, respectively. The numbers in the highlighted twelve mutants in the 11th column (lower part) were from a separate experiment. The ratio of pChk1/total Chk1 in wild-type control for the twelve mutants is 43.1 ± 4.7 (n = 3). The six primary mutants selected for further characterization are marked by the dots on the left. The two mutants with confirmed partial DRC defects are marked by the green dots. The red dots indicate the mutants whose "checkpoint defects" are caused by secondary mutations. Brown dots are those with largely intact checkpoints.

phosphorylation of Mrc1 and Cds1, particularly Cds1, was reduced in the #24 mutant (S5A and S5B Fig) although Chk1 phosphorylation was not reduced much in the presence of MMS (S5C and S5D Fig). These results suggest that the #24 mutant might have a significant defect in the DRC, not the DDC pathway. In addition, the specific Western blot in S5C Fig showed that the #17 mutant might be defective in Chk1 phosphorylation, although the quantitation results from the repeated experiments suggest otherwise.

In total, the two rounds of genetic screen have identified twenty-five *ssb1* primary mutants. A detailed summary of all twenty-five mutants in terms of the amino acid changes, cell growth, drug sensitivities, and checkpoint signaling defects are summarized in Fig 5. As shown in Fig 5A, the identified mutations are distributed across the entire molecule and more importantly, ~31% of amino acids within the N-terminal region have been mutated, suggesting that the screen is extensive or near exhaustion for identifying the non-lethal *ssb1* mutants with defects in checkpoint signaling or other functions (see Discussion).

## Secondary mutations in the #7 and the #24 primary mutants

So far, our extensive mutational analysis has identified a number of non-lethal *ssb1*mutants such as #1, #7, #17, and #24 mutants that might be defective in checkpoint signaling (Fig 5B). These four mutants as well as the #10 and the #19 mutants that showed the checkpoint defect to a lesser degree (Figs 4C and S5B) were then selected for further characterization (Fig 5B, marked by dots). We first integrate the *ssb1* mutations identified in the six mutants at the genomic locus using the method diagramed in S2E Fig. Tetrad dissections confirmed the single integration at *ssb1* for all six mutants (S6 Fig). These mutant integrants were hereafter renamed as *ssb1-1*, *ssb1-7*, *ssb1-10*, *ssb1-17*, *ssb1-19*, and *ssb1-24*. When drug sensitivities of the six integrants were compared with their primary mutants (Fig 6A), we found that *ssb1-1*, *ssb1-10*, *ssb1-17*, and *ssb1-19* showed similar sensitivities as their primary mutants, the *ssb1-24* integrant was less sensitive to both HU and MMS. Although *ssb1-7* showed a similar sensitivity to HU, it was less sensitive to MMS than its primary mutant. These results show that the #7 and the #24 primary mutants likely carry a secondary mutation. This is a surprise as the mutations were generated by precise allele replacement, and the primary mutants have been backcrossed with wild-type *S. pombe* at least once. We believe that the near-saturation screen is a contribting factor of the secondary mutations. Nevertheless, since the #24 mutant showed the most prominent checkpoint defect among the twenty-five mutants, we decided to investigate this mutant further. We first tagged the *ssb1* in the #24 mutant with a C-terminal HA linked with the *ura4* marker (S7A Fig). The tagged strain was backcrossed with wild-type *S. pombe* followed by tetrad dissection (S7B Fig). The 2:2 ratios of the *ura+* spores and the anti-HA Western analysis confirmed the tagging (S7C Fig). However, the *hus* (HU sensitive) phenotype varied among the *ura+* spores and some *ura-* spores also showed the *hus* phenotype (S7B Fig). When individually analyzed, all *hus* spores expressed the wild-type level Ssb1 (S7C Fig), showing that the varied *hus* phenotype is unrelated to Ssb1 levels. Using spot assay, we found while all *ura+* colonies were sensitive to both HU and MMS, the *ura-* spores were sensitive to HU but not MMS (S7D Fig), indicating that the #24 mutant carries a secondary mutation similar to the metabolic mutants we have previously reported [50,51]. When the cell cycle progression of the #24 mutant was analysed (S7E Fig), we found that a large fraction of the cell population was arrested by HU at G2/M, not the S phase. The #24 mutant lacking the secondary mutation was also found quite resistant to the acute HU treatment (S7F Fig). We conclude that the #24 mutant carries an unknown metabolic mutation that causes the "DRC defect" indirectly by the cell cycle effect. The #7 primary mutant, although not further investigated, likely carries a secondary mutation in DNA repair as the mutation sensitizes the cells to MMS, not HU (Fig 6A).

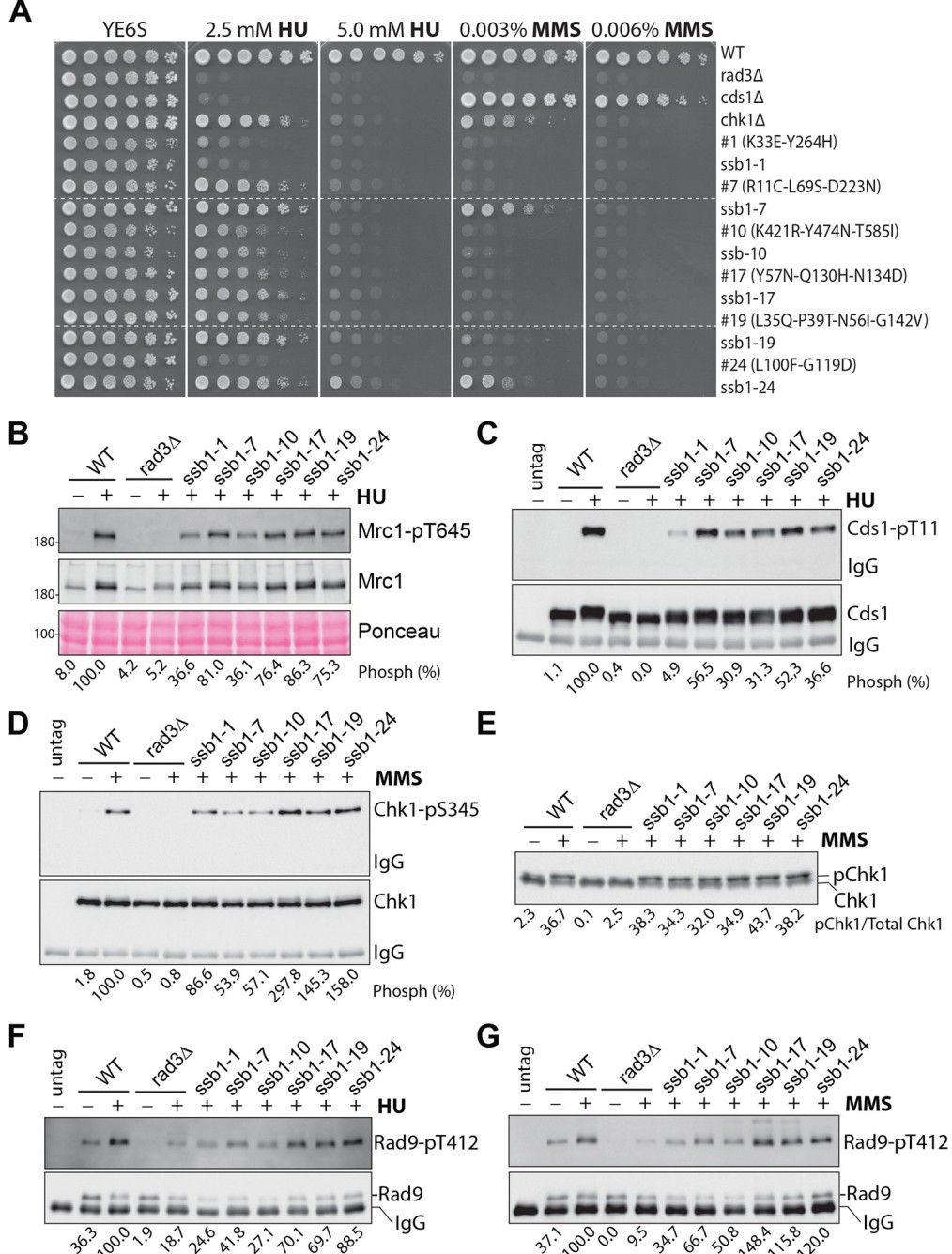

**Fig 6. Further characterization of the checkpoint defect in six *ssb1* mutants.** (**A**) Six primary *ssb1* mutants with more prominent checkpoint defects were selected. Their mutations were confirmed by integrating at the genomic locus in a wild-type strain. Drug sensitivities of the integrants referred to as *ssb1-1*, *ssb1-7*, *ssb1-10*, *ssb1-17*, *ssb1-19*, and *ssb1-24* were examined by spot assay and compared with their corresponding primary mutants. Dashed lines indicate the discontinuity. Phosphorylation of Mrc1 (**B**) and Cds1 (**C**) in the six mutant integrants was examined as in Fig 1B and 1D. Quantitation results from three independent blots are shown in S8A and S8B Fig, respectively. Chk1 phosphorylation in the six integrants was examined as in Fig 1F and 1H by phospho-specific antibody (**D**) and by mobility shift assay (**E**) and the quantitation results are shown in S8C and S8D Fig, respectively. Rad9 phosphorylation was examined in IPed samples using the phospho-specific antibody in the presence of HU (**F**) or MMS (**G**). Quantitation results are shown in S8E and S8F Fig, respectively.

## Partial DRC signaling defect in *ssb1-1* and *ssb1-10*

We then examined the checkpoint signaling defects in the six *ssb1* mutants whose mutations have been confirmed by the genomic integration. When phosphorylation of Mrc1 was examined in the presence of HU, we found that the phosphorylation was reduced in *ssb1-1* and *ssb1-10* and unaffected in *ssb1-7*, *ssb1-17*, *ssb1-19*, and *ssb1-24* mutants (Figs 6B and S8A). When Cds1 phosphorylation was examined, we found that it was more significantly reduced in *ssb1-1* and *ssb1-10* than the rest four mutants (Figs 6C and S8B). When Chk1 phosphorylation was examined by the phospho-specific antibody and the mobility shift assay, we found that the six mutants showed either increased (*ssb1-1*, *ssb1-17*, *ssb1-19*, and *ssb1-24*) or slightly reduced phosphorylation (*ssb1-7* and *ssb1-10*), suggesting a functional DDC (Figs 6D and 6E and S8C and S8D). We believe that the significant reduction of Chk1 phosphorylation in the #10 primary mutant as detected by the phospho-specific antibody (Fig 4A), but not the mobility shift assay (Fig 4D), is likely due to the technical issues during the primary screen (see Discussion). In the presence of DNA damage or replication stress, Rad9 of the 911 complex is phosphorylated by Rad3 to promote Chk1 and Cds1 activation although Tel1 also contributes to the phosphorylation at a basal level. We then examined Rad9 phosphorylation using a phospho-specific antibody for Rad9-pT412 [43,58]. In the presence of HU, Rad9 phosphorylation was reduced in *ssb1-1*, *ssb1-7*, and *ssb1-10* while it was moderately reduced in *ssb1-17*, *ssb1-19*, and *ssb1-24* mutants (Figs 6F and S8E). When treated with MMS, *ssb1-1*, *ssb1-7*, and *ssb1-10* showed a moderately reduced Rad9 phosphorylation, whereas the phosphorylation in *ssb1-17*, *ssb1-19*, and *ssb1-24* was at the wild-type level or slightly higher (Figs 6G and S8F). Together, these results suggest that while *ssb1-7* has a minor defect in the DRC, *ssb1-1* and *ssb1-10* have a more severe DRC defect. On the other hand, the checkpoints in *ssb1-17*, *ssb1-19*, and *ssb1-24* are largely normal or minimally compromised.

## Further evidence of the DRC defect in *ssb1-1* and *ssb1-10*

To confirm the DRC defect in *ssb1-1* and *ssb1-10*, particularly *ssb1-1*, we first examined the sensitivities of the mutants to acute treatment with HU and MMS by spot assay (Fig 7A). The results showed that while *ssb1-1* and *ssb1-10* were slightly sensitive to HU, the *ssb1-7*, *ssb1-17*, *ssb1-19*, and *ssb1-24* were relatively insensitive. To confirm the acute HU sensitivity, we performed the colony recovery assay (Fig 7B) and found that while *ssb1-7* was insensitive, *ssb1-1* and *ssb1-10* were sensitive although the sensitivities were much lower than *cds1Δ* cells, consistent with the partial DRC defect. All six mutants were highly sensitive to acute MMS treatment (Fig 7A). Interestingly, except *ssb1-7*, the acute MMS sensitivities were even higher than cells lacking Chk1, suggesting a defect in DNA repair (see below).

Both *ssb1-1* and *ssb1-10* mutants have a growth defect, as evidenced by their different and overall smaller sizes of colonies of (S9A Fig), which may indirectly affect the DRC. To preclude this possibility, we examined Mrc1 phosphorylation every hour during the HU treatment (Figs 7C and S9B). Unlike the *ssb1-7* mutant in which Mrc1 phosphorylation was slightly reduced during the six hours of HU treatment, the phosphorylation was significantly reduced in *ssb1-1* and *ssb1-10* cells, particularly during the first three hours of HU treatment. When the cell cycle progression was monitored in the presence of HU by flow cytometry, most of the wild-type and *rad3Δ* cells were arrested at the S phase in ~3 h (Fig 7D). However, unlike wild-type cells that finished the bulk of DNA synthesis in ~7 or 8 h in HU, *rad3Δ* cells failed to continue the DNA synthesis in HU. Under similar conditions, *ssb1-1*, *ssb1-7*, and *ssb1-10* mutants were all arrested in the S phase in ~3 h, unlike the G2/M arrest observed in the #24 primary mutant (S7E Fig). Furthermore, *ssb1-7* cells finished the bulk DNA synthesis almost like the wild-type cells, whereas the DNA synthesis in *ssb1-1* and *ssb1-10*, particularly *ssb1-1*, was slightly slower

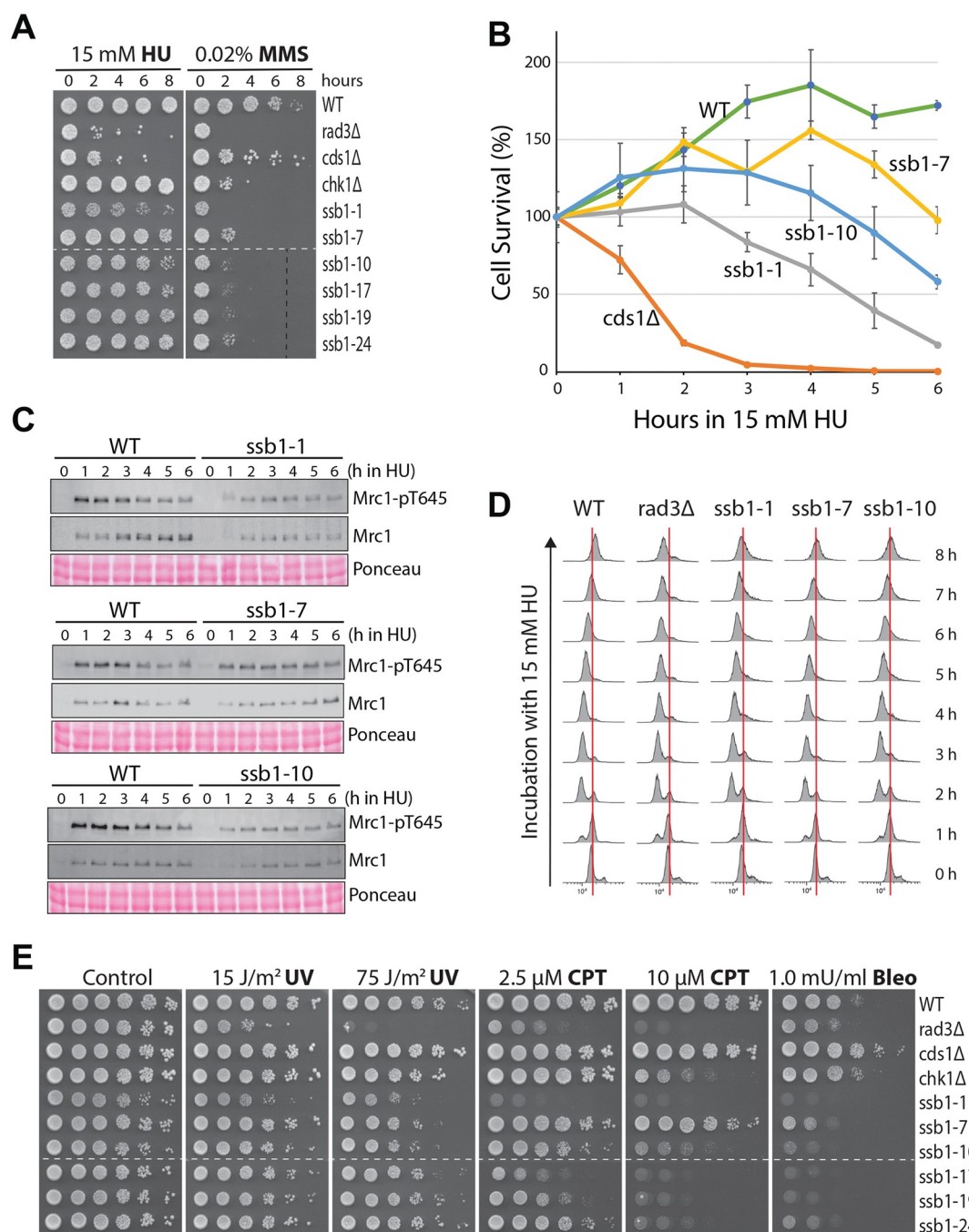

**Fig 7. Defects in checkpoint signaling and DNA repair in *ssb1-1* and *ssb1-10*.** (**A**) Sensitivities of the six *ssb1* mutants to acute treatment with HU (left) and MMS (right) were determined by spot assay. Dashed lines: discontinuity. (**B**) Acute HU sensitivities of *ssb1-1*, *ssb1-7*, and *ssb1-10* were determined by colony recovery assay. (**C**) Time-course analysis of Mrc1 phosphorylation in *ssb1-1*, *ssb1-7*, and *ssb1-10* in the presence of 15 mM HU. The quantitation results are shown in S9B Fig. (**D**) Cell cycle progression of *ssb1-1*, *ssb1-7*, and *ssb1-10* in the presence of 15 mM HU was analyzed by flow cytometry. Red lines indicate 2C DNA contents. (**E**) Sensitivities of the six *ssb1* mutants to UV, CPT, and bleomycin were determined by spot assay. Dashed lines indicate discontinuity.

than in wild-type cells. These results exclude the possibility of the G2/M arrest by HU like in #24 primary mutant (S7E Fig) and support the conclusion that the DRC is partially defective in *ssb1-1* and *ssb1-10*.

When treated with HU, the DRC mutants undergo premature mitosis, generating a so-called *cut* (cells untimely torn) phenotype [59] in *S. pombe* that can be examined under microscope after staining the cells with Hoechst for genomic DNA and Blankophor for septum. As shown in S10 Fig, after the HU treatment for six hours in liquid cultures, wild-type cells were elongated and mononuclear whereas the *rad3Δ* cells were all short and most of the cells showed the *cut* phenotype (arrows). Unlike the *rad3Δ* cells, the *cds1Δ* cells were elongated in HU because the DDC remains activated in the presence of collapsed forks. However, >30% *of cds1Δ* cells showed the *cut* phenotype due to the lack of DRC (S10 Fig, bottom right). In contrast, *chk1Δ* cells were elongated and only 4.6% (±1.2, n = 3) cells showed the *cut* phenotype in HU, consistent with its main function in the DDC, not the DRC pathway. Under similar conditions, all six mutants were found elongated in HU. However, more *cut* cells were observed than in wild-type cells except *ssb1-17*. Among the six mutants, *ssb1-1* showed the highest number of *cut* cells although the number was much smaller than in *cds1Δ* cells (S10 Fig, lower right panel). On the other hand, the number of *cut* cells in *ssb1-10* was similar to *chk1Δ* cells. These microscopic data further support the partial DRC signaling defect in the *ssb1-1* and *ssb1-10* mutants.

## Defects in DNA repair

The significant acute MMS sensitivities of the six mutants suggest a defect in DNA repair (Fig 7A). To further investigate, the sensitivities of the six mutants to ultraviolet (UV) irradiation, camptothecin (CPT), and bleomycin were examined by spot assay (Fig 7E). While UV directly generates pyrimidine dimers in DNA [60], CPT stabilizes covalent DNA-topoisomerase I complex, which when encountered with replication fork, can be converted into one-ended double-strand break [61]. Bleomycin cleaves DNA, generating strand breaks [62]. We found that all six mutants were sensitive to UV, and the sensitivities were lower than *rad3Δ* but higher than *cds1Δ* and *chk1Δ* cells (Fig 7E). The six mutants, except *ssb1-7*, were highly sensitive to CPT and the sensitivity was comparable to that in *chk1Δ*. Remarkably, all mutants were more sensitive to bleomycin than *rad3Δ* cells that lack checkpoints. These results strongly indicate that the six *ssb1* mutants, including *ssb1-1* and *ssb1-10*, are defective in DNA repair, particularly the pathways for repairing strand breaks.

## Discussion

To better understand the checkpoint signaling function of RPA in cells, we have carried out an genetic screen in fission yeast to identify mutants of the large subunit Ssb1. This extensive screen has identified twenty-five primary mutants that are sensitive to HU and MMS. Preliminary studies of the primary mutants uncovered six mutants with defects in Rad3 kinase signaling. Among the six primary mutants, the checkpoint defect in the #24 mutant is the most prominent. Further characterization of the six checkpoint defective mutants showed that although the mutations were generated by precise allele replacement at the *ssb1* genomic locus, two of the six primary mutants, including the #24 mutant, carry a secondary mutation. The secondary mutation in the #24 mutant sensitizes the cells to HU, but not MMS, which behaves like the metabolic mutants we have previously identified during our genome-wide *hus* screen [50,51]. Since HU arrests the metabolic mutants in G2/M, not the S phase, the observed "DRC defect" in the #24 primary mutant is likely caused indirectly by the cell cycle effect. Nevertheless, after confirming the mutations by genomic integration followed by tetrad dissection and

checkpoint analysis, our comprehensive mutational analysis has identified two mutants *ssb1-1* and *ssb1-10* that show a partial Rad3 signaling defect primarily in the DRC, not the DDC pathway.

Several pieces of evidence support the partial DRC defect in *ssb1-1* and *ssb1-10*. First, Rad3-dependent phosphorylations of Rad9, Mrc1, and Cds1 are all reduced in HU- or MMS-treated cells (Fig 6B–6G) and the reduced phosphorylations are not due to the indirect cell cycle effect (Fig 7D). Second, consistent with the reduced Rad3 phosphorylations in the DRC pathway, the two mutants show moderate to minimal sensitivities to acute treatment with HU (Fig 7A and 7B). Third, although the numbers are low, the two mutants show *cut* cells in the presence of HU (S10 Fig). The low numbers of *cut* cells and the cell elongation in HU observed in the two mutants are likely due to their functional DDC pathway because Chk1 phosphorylation is largely unaffected (Figs 6D and 6E and S8C and S8D). Finally, the protein levels of Ssb1 are normal or slightly increased in the two mutants (Fig 3D) although they all show a growth defect (S9A Fig). We believe that the growth defect of the two mutants is unrelated to their partial DRC defect, and the time course analysis of Mrc1 phosphorylation (Fig 7C) and the flow cytometry data (Fig 7D) support this conclusion.

Although the two *ssb1* mutants with only a partial checkpoint defect in the DRC pathway are identified, this targeted screen is likely extensive or near exhaustion, because (1) the mutated residues in the two previously reported *ssb1* mutants *rad11A (ssb1-R339H)* and *ssb1-D223Y* were identified at least once by the screen (Fig 5A). The Gly$^{78}$ residue in *ssb1-G78E* was not identified likely due to its moderate sensitivities to HU and MMS (Fig 1A). (2) >40% of the mutations were individually identified at least two times (Fig 5B). And (3) ~31% amino acid residues in the N-terminal region and ~4.4% in the rest of the Ssb1 molecule were mutated (Fig 5A). We believe that our screened *ssb1-1* and *ssb1-10* mutants, particularly the former, have maximally eliminated the checkpoint function that can be genetically separated in Ssb1. The partial checkpoint defect can be explained by at least three possibilities. First, the amino acid residues that function in checkpoint signaling in Ssb1 are also required for cell survival. Second, although this screen focuses on Ssb1, Ssb2 may also contribute to checkpoint signaling. Finally, RPA may function redundantly with an unknown factor in Rad3-mediated checkpoint initiation in fission yeast.

The results in Fig 7E indicate that *ssb1-1* and *ssb1-10* are also defective in strand break repair, which echoes the major homologous recombination defect in the budding yeast *rfa1-t11* mutant [28,29,35]. The rest four *ssb1* mutants are also more sensitive to bleomycin than *rad3Δ* cells, suggesting an important role of Ssb1 in strand break repair. Surprisingly, none of the identified mutants show a significant defect in Chk1 phosphorylation of the DDC pathway because strand break repair mainly occurs at G2 where the DDC is highly functional. Since more repair mutants were identified than the checkpoint mutants by this screen, it is possible that the repair function of Ssb1 can be more readily separated genetically from its essential function or it plays an more important role in DNA break repair than the checkpoint. The specific DRC defect in *ssb1-1* and *ssb1-10* described here is similar to the S phase checkpoint defect of the budding yeast mutant *rfa1-M2* [30], but not the DNA damage checkpoint defect in *rfa1-t11* [31,35]. As mentioned above, Chk1 phosphorylation in the DDC pathway is commonly monitored by mobility shift assay. Using this assay, we found that all twenty-five primary *ssb1* mutants did not show a significant defect in Chk1 phosphorylation. Since Rad3 also phosphorylates other residues on Chk1, we were concerned with the non-essential phosphorylation events that might affect the sensitivity of the mobility shift assay leading to a wrong conclusion. To eliminate this concern, we generated a phospho-specific antibody for phosphorylated Chk1-Ser$^{345}$. Although the antibody is highly specific and sensitive, we found that Western blottings using the antibody show significant variations among experimental

repeats. Nonetheless, although the differences are noticeable, the experimental results obtained by using the phospho-specific antibody are generally consistent with that from the mobility shift assay. Some of the differences in the results with the primary mutants by the two methods are likely technical as mentioned above or due to the complication of secondary mutations. Indeed, after the secondary mutation was removed from the #24 primary mutant, Chk1 phosphorylation was increased in *ssb1-24* to a level similar to or higher than in wild-type cells (compare S5C and S5D with S8C and S8D Figs). We believe that two assays used here are sensitive enough to detect a minor defect in Chk1 phosphorylation and the largely functional DDC observed in the *ssb1* mutants reflects the real situation inside the cells. The cell elongation in HU-treated *ssb1* mutants including *ssb1-1* and *ssb1-10* (S10 Fig) further supports the conclusion.

There are two missense mutations in *ssb1-1*. The first one causes a charge reversal substitution of $Lys^{33}$ with glutamic acid whereas the second mutation substitutes $Tyr^{264}$ with histidine. $Lys^{33}$ is within the N-terminal DBD-F (S11 Fig), which suggests that according to the current model, it may function in recruiting Rad26, the ATRIP homolog in fission yeast. The *ssb1-10* mutant, although less defective in the DRC, has three mutations that are all in or near the C-terminal DBD-B and -C, suggesting that these mutations may affect the Rad3 kinase signaling through a different mechanism. Our preliminary results show that both mutations in *ssb1-24 (L100F-G119D)* contribute the drug sensitivities. However, it remains possible that by-stander mutations exist in the mutants with multiple mutations identified by this intensive screen. Nevertheless, further studies are needed to investigate the genetic interactions of the two *ssb1* mutants with other checkpoint mutants and their defects in physical interactions with other checkpoint proteins. The partial checkpoint defect of *ssb1-1* and *ssb1-10* also suggests a redundant factor in checkpoint initiation. Similar to the two *ssb1* mutants, our previous *hus* screen has identified several mutants that are defective more specifically in the DRC pathway [39,40,41]. It would be interesting to investigate how the *ssb1* mutants interact genetically with those *hus* mutants. Together, the genetic data from both yeasts strongly suggest that the molecular mechanisms by which ATR initiates the checkpoint signaling, particularly at the replication fork, remain to be fully understood.

The remaining nineteen *ssb1* primary mutants show a minimal or uncompromised checkpoint defect and are likely defective in other cellular processes. Their sensitivities to various DNA-damaging agents strongly suggest that at least some of them are defective in DNA repair. Indeed, as mentioned above, the *ssb1-1*, *ssb1-10*, *ssb1-17*, *ssb1-19*, and *ssb1-24* mutants are more sensitive to strand breaks induced by bleomycin than *rad3Δ* cells (Fig 7E). Preliminary data have also shown significantly shorter or complete loss of telomeres in some of the *ssb1* primary mutants, which support its important role in telomere maintenance (38). Further studies are needed to eliminate the secondary mutations from the remaining nineteen mutants and dissect the versatile functions of RPA in genome maintenance. The previously uncharacterized *ssb1* mutants described in this study provide a valuable tool for future investigations in fission yeast.

## Materials and methods

### Yeast strains and plasmids

*S. pombe* strains were cultured at 30°C in YE6S (0.5% yeast extract, 3% dextrose, and 6 supplements) or synthetic EMM6S medium lacking the appropriate supplements [63]. Yeast strains, plasmids, and PCR primers used in this study are listed in Supplementary Table S1, S2, and S3, respectively. Mutations were identified by DNA sequencing (Retrogen, San Diego, CA).

## The genetic screen of *ssb1* mutants

The *ssb1* mutants were screened by the targeted forward genetic approach [55]. The *ssb1* expression cassette was cloned into the *S. pombe* pJK210 integrating vector that carries the *ura4* marker [56] (see S3A Fig). To facilitate the cloning, NdeI, and XmaI sites were engineered into the vector before and after the ORF, respectively. A SpeI restriction site was also introduced in the middle of the ORF by a silent mutation for the convenience of making random mutations. The random mutations were made by mutational PCR [57] of the 5'-terminal and the 3'-terminal halves of *ssb1* in two separate libraries. To integrate the library with mutations in the C-terminal half for precise allele replacement, the library was linearized with NheI. Similarly, the N-terminal mutational library was linearized with BstBI (see diagram in Fig 3A). For screening the N-terminus 154 amino acid region, random mutations were made by PCR between NdeI and PstI sites. The linearized library DNA was transformed into wild-type *S. pombe* lacking *ura4*. The transformed cells were selected by sequential cultures in EMM6S [ura-] medium during the first pop-in step (S3A Fig, step 1 and 2). In the next pop-out step, the cells with integrated *ura4* marker were grown up in 150 ml YE6S media to saturation to lose the *ura4* marker. The cells were then spread onto 5-FOA plates to counter-select the cells that had lost the *ura4* gene. The colonies formed on 5-FOA plates were replicated onto YE6S plates containing 5 mM HU. The colonies with *hus* phenotype were selected, streaked out into single colonies, and the sensitivities to HU and MMS were assessed by spot assay. The selected mutants were backcrossed at least once before DNA sequencing to identify the mutations. The backcrossed mutants were also used for analyzing drug sensitivities (Figs 3A and S4A) and checkpoint signaling defects (Figs 4 and S5).

## Integration of *ssb1* mutations at the genomic locus

*ssb1* with the identified mutations were cloned into a integration vector with a *kanR* gene (Table S2, pYJ1827, pYJ1919-1923). The plasmids were digested with BglII and NcoI to purify the DNA fragment to replace the wild-type allele at the genomic locus by a marker switching method diagrammed in S2E Fig. Integration of the *ssb1-K45E*, *ssb1-R46E*, and *ssb1-K45E-R46E* mutations (desginated to be equivalent of *S. cerevisiae rfc1-t11*) in *S. pombe* was also carried out by the marker switching method.

## Drug sensitivity assay

Sensitivities to HU and the DNA damaging agents were determined by spot assay or in liquid cultures as described in our previous studies [50,51]. For the spot assay to assess the acute HU and MMS sensitivity, logarithmically growing cells were diluted to $2 \times 10^6$/ml in 10 ml YE6S medium. After 15 mM HU or 0.02% MMS was added to the culture, the cells were incubated at 30˚C. Every hour during the drug treatment, an equal amount of the culture was removed. The cells were collected by centrifugation, washed once, diluted 10-fold in dH$_2$O, and spotted on YE6S plates for cell recovery. The plates were incubated at 30˚C for 3 days before being photographed.

## Immunopurification

$1 \times 10^8$ logarithmically growing cells were harvested and saved at -20˚C in a 1.5 ml screw cap tube. The frozen cell pellets were lysed by a mini-bead beater in the buffer containing 25 mM HEPES/NaOH (pH 7.5), 50 mM NaF, 1 mM NaVO$_4$, 10 mM NaP$_2$O$_7$, 40 mM ß-glycerophosphate, 0.1% Tween 20, 0.5% NP-40, and protease inhibitors. The lysates were centrifuged at 16,000 g, 4˚C for 5 min to make the cell extract. Cell extract was incubated with prewashed

antibody agarose resin by rotating in 2 ml tubes at 4˚C for 2 h. The resins were washed three times with TBS-T at 4˚C for 20 min. The IPed samples were separated by SDS-PAGE followed by Western blotting.

## Western blotting

Analyses of phosphorylated Rad9-Thr[412], Mrc1-Thr[645], and Cds1-Thr[11] by Western blotting using the phospho-specific antibodies have been described in our previous studies [43,44,45]. The custom antibody against phosphorylated Chk1-Ser[345] used in this study was generated in rabbits and purified by Bethyl Laboratories (Montgomery, TX). The chemically synthesized peptide VYGAL**pS**QPVQL was used as the immunogen. The specificity of this antibody is confirmed by Western blotting (S1 Fig). For Western blotting using the Chk1-pS345 antibody, Chk1-3HA or Chk1-9myc2HA6his was IPed with anti-HA antibody beads (sc-7392AC, Santa Cruz Biotech., TX) from the whole cell lysates made by a mini-bead beater as described above in the cell lysis buffer containing 150 mM NaCl. The IPed sample was analyzed on an 8% SDS-PAGE gel. After transfer to a nitrocellulose membrane, the membrane was blotted with the Chk1-pS345 antibody at the 1:3000 dilution for 3 h to detect the phosphorylated Chk1-Ser[345] in ChemiDoc (Bio-Rad). The membrane was stripped, extensively washed for $\geq$ 3 h, and then reblotted with an anti-HA antibody (clone 12CA5, Sigma) to reveal Chk1. The band intensity was quantified using Image Lab (Bio-Rad). After normalizing with the Chk1 signal, the intensity of the Chk1-pS345 band is shown in percentages as compared with MMS-treated wild-type cells. To examine Chk1 phosphorylation by mobility shift assay, the whole cell lysate made from TCA-fixed cells was separated by an 8% SDS-PAGE gel. After transferring to a nitrocellulose membrane, both Chk1 and phosphorylated Chk1 were detected using the anti-HA antibody. The intensities of the two bands were quantitated. The ratio of phosphorylated Chk1 vs total Chk1 is shown in percentages.

For generating the custom antibody against Ssb1, Ssb1 was tagged with 6xhis at the N-terminus in pBG100 vector and expressed in BL21(DE3) cells. To start the protein expression, 0.4 mM IPTG was added to a 2 L culture of logarithmically growing *E. coli* at 37˚C. The culture was continued at 37˚C for 3 h. The cells were harvested and resuspended in 20 mM phosphate buffer containing 250 mM NaCl and 20 mM imidazole (pH 8.0). The cells were lysed by running through a cell disruptor (Avestin, Inc) three times at 4˚C. The cell lysate was centrifuged at 43,000 g, 4˚C, for 20 min. After removing the supernatant completely, the pellet was dissolved in the lysing buffer containing 8 M urea. After clarification by centrifugation, the supernatant was loaded onto a 5 ml column with prewashed Talon resin (Clontech Laboratories, Inc, CA). The column was washed three times with the low pH buffer (pH 6.3), and the Ssb1 protein was eluted in the phosphate buffer (pH 7.5) containing 300 mM imidazole and 2.4 M urea. The eluted sample was concentrated in Amicon (Millipore) and used as the immunogen in rabbits (Colcalico Biologicals, Inc, PA). The specificity of the antibody was verified by Western blotting of the whole *S. pombe* cell lysate (Fig 2B).

## Flow cytometry

$1 \times 10^7$ logarithmically growing cells were collected, fixed in 70% ethanol, and analyzed by Accuri C6 flow cytometer as described in our previous studies [50,51].

## Microscopy

The cells were fixed onto glass slides by heating at 75˚C for ~30 sec. The cells fixed on the slides were stained in PBS buffer containing 5 µg/ml Hoechst33258 (Sigma-Aldrich) and 1:100 dilution of the Blankophor working solution (MP Biochemicals). The stained cells were examined

using an Olympus EX41 fluorescent microscope. Images were captured with an IQCAM camera (Fast1394) using Qcapture Pro 6.0 software and then extracted into Photoshop (Adobe) to generate the S10 Fig.

## Supporting information

**S1 Fig. Specificity of the phospho-specific antibody against Chk1-pS345.** Logarithmically growing cells were treated with 0.01% MMS in YE6S medium for 90 min at 30˚C. 5.0 OD cells were harvested from each culture and saved in a screw-cap microtube at -20˚C. The frozen cell pellets were lysed by mini-bead beater in HEPES/NaOH buffer containing 150 mM NaCl and inhibitors of phosphatases and proteases. Chk1-HA was IPed using anti-HA antibody beads in cold room for 2 h. The IPed samples were separated on an 8% SDS-PAGE gel and then transferred to a nitrocellulose membrane. The membrane was strained with Ponceau-S to show the IgG bands between the 60 and 45 KDa markers (lower panel). The membrane was first incubated with the phospho-specific antibody at 1:3000 dilutions for 3 h at room temperature to reveal phosphorylated Chk1(top panel). After stripping, the membrane was extensively washed in deionized water and then reblotted with anti-HA antibody to reveal the HA-tagged Chk1 (middle panel). NW223 strain expressing HA tagged Chk1 was used as the wild-type cells, whereas NW444 expressing HA tagged Chk1-S345A was used as the mutant control. TK7 STRAIN expressing untagged Chk1 was used as the control for specific IP.
(TIF)

**S2 Fig. Integration of *S. cerevisiae rfa1-t11* mutation significantly suppresses cell growth in *S. pombe*.** (**A**) Diagram of Ssb1, the F domain, and the *rfa1-t11* mutation. The positively charged Lys$^{45}$ residue mutated in *rfa1-t11* is highly conserved from yeasts to humans. Sc: *S cerevisiae*, Sp: *S. pombe*, Dm: *Drosophila melanogaster*; Xl: *Xenopus laevis*, Mm: *Mus musculus*, and Hs: *Homo sapiens*. (**B**) Strategy for tagging *ssb1* with an HA epitope linked with *ura4* marker at the genomic locus. The resulting *ura*$^+$ colonies were selected by colony PCR to confirm the correct 5' and 3' integrations. nmtT: *nmt* terminator. (**C**) The tagged strain was confirmed by tetrad dissection that showed 2:2 ratios of *ura*$^+$ and *ura*$^-$ spores for all dissected asci. Dashed line indicates discontinuity. (**D**) The tagged strain was also confirmed by Western blotting of the whole cell lysates using anti-HA antibody. (**E**) The marker switching method for replacing wild type *ssb1* at the genomic locus with the mutant *ssb1-R46E*. The *ssb1-K45E* mutation is likely lethal in *S. pombe* as its replacement did not generate any colonies (data not shown). (**F**) The *ssb1-R46E* mutant showed a severe growth defect on YE6S plate.
(TIF)

**S3 Fig. Genetic screen of new *ssb1* mutants by precise replacement with alleles of random mutations at the genomic locus.** (**A**) The strategy for *ssb1* replacement (55). The ORF of *ssb1* was mutated by PCR at the N- and C-terminal halves separately to generate two libraries. After linearization by enzyme digestion, the library DNA is transformed into wild-type *S. pombe* lacking the *ura4* gene. The cells were cultured in EMM6S[ura-] to select the transformants with the integrated *ura4* marker. The *ura4*$^+$ cells were then cultured in YE6S to pop-out the *ura4* marker to be counter selected by 5-FOA. The colonies formed on 5-FOA plates carry either wild type or mutant *ssb1* at the genomic locus. The *ssb1* mutants were screened by replica plating on HU plates. (**B**) The screened mutants were streaked out into single colonies for confirming the drug sensitivities. As an example, the HU and MMS sensitivities of the primary mutants screened with N-terminal half library were assessed by three-spot assay. The drug sensitive mutants marked by red asterisks were backcrossed once, renamed, and then saved for further investigation. (**C**) The *ts* phenotype of some of the screened mutants was assessed by

spot assay. The *tel2-C307Y mutant*, used as a control, is a *ts* mutant that we screened previously
(39).
(TIF)

**S4 Fig. Twelve primary *ssb1* mutants were identified by 2<sup>nd</sup> round screening of the N-terminal 154 amino acid region.** (**A**) Sensitivities of the twelve mutants (#14 - #25) to HU and
MMS were examined by spot assay. The amino acid changes in the mutants are shown on the
right. Dashed lines indicate discontinuity. (**B**) Acute HU sensitivity of the twelve *ssb1* mutants
was assessed by spot assay. Dashed lines indicate discontinuity. (**C**) Ssb1 protein levels were
examined in logarithmically growing wild-type and the mutant cells by Western using anti-
Ssb1 antibody (top panels). A section of Ponceau S-stained membrane is shown for loading.
The Western blotting was repeated three times. Quantitation results are shown (lower panel).
Error bars: means and SDs of the triplicates.
(TIF)

**S5 Fig. Checkpoint signaling defects in the twelve *ssb1* mutants identified by screening the
N-terminal 154 amino acid region.** (**A**) Mrc1 phosphorylation by Rad3 was examined by
Western blotting before (-) or after (+) the cells were treated with 15 mM HU for 3 h (left).
Quantitation results are shown on the right. Error bars are the means and SDs of three inde
pendent blots. (**B**) Cds1 phosphorylation by Rad3 was examined by Western blotting in the
mutants. Quantitation results are shown on the right. (**C**) Chk1 phosphorylation by Rad3 was
examined by Western blotting using phospho-specific antibody against Chk1-pS345 after the
cells were treated with 0.01% MMS for 90 min (left). Quantitation results are shown on the
right. (**C**) Chk1 phosphorylation was also examined by the commonly used mobility shift
assay. Wild-type and the mutant cells were treated with 0.01% MMS for 90 min. The cell
lysates made by TCA method were analysed by an 8% SDS PAGE gel for Western blotting
with anti-HA antibody. Quantitation results are shown on the right.
(TIF)

**S6 Fig. Confirmation of the integration of the *ssb1* mutations identified in six selected pri
mary mutants at the genomic locus by tetrad dissection.** The untagged *ssb1* integrants with
*ssb1-1*, *ssb1-7*, *ssb1-10*, *ssb1-17*, *ssb1-19*, and *ssb1-24* mutations linked to the kanR marker were
made by using the marker switching method shown in S2E Fig. The integrants were back
crossed with the wild type LLD3427 strain carrying a *ura4* marker or TK7 lacking the *ura4*
marker. Tetrad dissections were performed for each cross and colonies formed on YE6S plates
were replicated onto plates containing 5 mM HU and the lethality dye phloxine B, YE6S plates
containing 100μg/ml G418, and EMM6S[ura-] plates. All tetrads showed 2:2 ratios of kanR or
ura<sup>+</sup> spores and the *hus* phenotype is absolutely linked to the kanR marker in all integrants.
Dashed lines: discontinuity. These results confirm single integration in the genome.
(TIF)

**S7 Fig. The observed DRC defect in #24 primary mutant is due to a secondary metabolic
mutation.** (**A**) Strategy for tagging the #24 mutant *ssb1* with HA linked with a *ura4* marker.
nmtT: *nmt* terminator. (**B**) The tagged strain YJ1836 was crossed with the wild-type TK7 strain
for tetrad dissection. Colonies were replicated onto HU and EMM6S[ura-] plates. This tetrad
dissection identified three groups of spores with the *hus* phenotype. Spores in the first group
are ura<sup>+</sup> with severe *hus* phenotype such as the 10a and 11c spores. Those in the second group
are ura<sup>+</sup> with a lower HU sensitivity such as the 7a and 8a spores. Spores in the third group are
ura<sup>-</sup> such as 1b and 11a. (**C**) Ssb1 levels in the representatives of the three groups were exam
ined and compared with wild type TK7 and YJ1836 cells. (**D**) HU and MMS sensitivities of
representative spores were assessed by spot assay. Note: the ura<sup>-</sup> spores in the third group are

resistant to MMS, suggesting a secondary unknown metabolic mutation in the #24 mutant (50, 51). (**D**) Consistent with the metabolic mutation, HU arrested a large fraction of the #24 mutant cells in G2/M, not S phase, which explains the observed "checkpoint defect" in the DRC. (**E**) After removing the secondary mutation, the #24 mutant became insensitive to acute HU treatment as determined by colony recovery assay. Data points are means of the numbers of recovered colonies on three separate plates.
(TIF)

**S8 Fig. Quantitation results of the checkpoint signaling defects in the six *ssb1* mutant integrants shown in Fig 6B–6G.** (**A**) Mrc1 phosphorylation in HU-treated wild type and the mutant cells was examined by Western blotting shown in Fig 6B. Quantitation results from three independent blots are shown. Error bars are means and SDs of the triplicates. (**B**) Quantitation results for Cds1 phosphorylation in Fig 6C from three repeats are shown. (**C**) Chk1 phosphorylation was examined in MMS-treated cells by Western blotting using the phospho-specific antibody as shown in Fig 6D. The quantitation results are shown from three repeats. (**D**) Quantitation results for Chk1 phosphorylation examined by mobility shift assay as shown in Fig 6E. (**E**) Rad9 phosphorylation in the 911 complex in HU-treated cells were examined as in Fig 6F. Quantitation results are from three separate blots. (**F**) Rad9 phosphorylation in MMS-treated cells were examined as in Fig 6G. Quantitation results are from three independent blots.
(TIF)

**S9 Fig. The cell growth defect and reduced Mrc1 phosphorylation in *ssb1-1* and *ssb1-10* mutants.** (**A**) Logarithmically growing wild type and the mutant cells were spread on YE6S plates. The plates were incubated at 30°C for 3 days to allow colony formation. Note: the size of *ssb1-1* and *ssb1-10* colonies varies significantly and is generally smaller than that of wild-type cells, showing a growth defect. (**B**) Time course of Mrc1 phosphorylation in HU-treated wild type (grey columns) and the mutant (black columns) cells were analysed by Western blotting as in Fig 7C. Error bars are means and SDs of triplicates.
(TIF)

**S10 Fig. Microscopic examination of the HU-treated *ssb1* mutant cells.** In the presence of HU, *ssb1-1* and *ssb1-10* mutants showed premature mitotic or *cut* cells in HU (red arrows) and the numbers of abnormal mitotic cells are higher than or similar to that in *chk1Δ* cells. Wild type, *rad3Δ*, *cds1Δ*, *chk1Δ*, and the six *ssb1* integrant mutant cells were treated with 15 mM HU for 6 h, double-stained with Hoechst and Blankophor, and then examined under the microscope. The *cut* cells were counted for wild-type, *cds1Δ*, *chk1Δ* and the six ssb1 mutants in a total of ≥ 150 cells for each sample, repeated the counting three times, and presented in percentages shown in the bottom right. Error bars represent the means and SDs of triplicates.
(TIF)

**S11 Fig. The AlphaFold structure of *S. pombe* Ssb1 and the relative positions of the five mutated resides identified in the *ssb1-1* and *ssb1-10* mutants.** Ssb1 Alphafold secondary structure (1–609 aa) was downloaded from the Pombase (https://www.pombase.org/gene/SPBC660.13c) and edited using PyMOL software [64]. The α-helices, β-sheets, and loops are colored in red, yellow, and green, respectively, in the four conserved DNA binding domains F, A, B, and C. The N- and C-termini are indicated by arrows. The mutated residues K33 and Y264 in *ssb1-1* are indicated by blue and the residues K421, Y474, and T585 in *ssb1-10* are shown in magenta.
(TIF)

**S1 Table. List of *S. pombe* strains used in this study.**
(DOC)

**S2 Table. List of plasmids used in this study.**
(DOCX)

**S3 Table. List of PCR and sequencing primers used in this study.**
(DOCX)

## Acknowledgments

We thank NBRP/YGRC in Japan and Dr. Nancy Walworth for providing the yeast strains. We also thank other members of the Xu lab for help and support.

## Author Contributions

**Data curation:** Yong-jie Xu.

**Funding acquisition:** Yong-jie Xu.

**Investigation:** Yong-jie Xu, Sankhadip Bhadra, Alaa Taha A. Mahdi, Toru M. Nakamura.

**Methodology:** Sankhadip Bhadra, Alaa Taha A. Mahdi, Ilknur Yurtsever, Toru M. Nakamura.

**Project administration:** Yong-jie Xu.

**Software:** Kamal Dev.

**Supervision:** Yong-jie Xu.

**Validation:** Alaa Taha A. Mahdi, Ilknur Yurtsever, Toru M. Nakamura.

**Visualization:** Kamal Dev.

**Writing – original draft:** Yong-jie Xu.

**Writing – review & editing:** Yong-jie Xu, Sankhadip Bhadra, Kamal Dev, Ilknur Yurtsever, Toru M. Nakamura.

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
