## [Decision Letter · Decision Letter 0]

3 Apr 2023

Dear Dr Xu,

Thank you very much for submitting your Research Article entitled 'Comprehensive mutational analysis of the checkpoint signaling functions of Rpa1/Ssb1 in fission yeast' to PLOS Genetics.

The manuscript was fully evaluated at the editorial level and by independent peer reviewers. The reviewers appreciated the attention to an important topic but identified some concerns that we ask you address in a revised manuscript. Most concerns can be addressed by changes to the text. There is an interesting suggestion from reviewer 2 that you map the mutations onto an alpha-fold generated structure, which could be very useful. Reviewer 3 has raised some concerns about Fig. 3, and about quantification in Fig. 5, that you should address. This reviewer has also suggested that you analyze the cut phenotype in an ssb1-1 chk1del background; you may wish to consider this, but it is not an essential requirement.

We therefore ask you to modify the manuscript according to the review recommendations.

Yours sincerely,

Geraldine Butler

Section Editor

PLOS Genetics

Reviewer's Responses to Questions

**Comments to the Authors:**

Reviewer #1: See the attachment.

Reviewer #2: This paper presents a comprehensive mutational analysis of the large subunit of RPA in fission yeast, in the effort to identify mutants that specifically disrupt the replication checkpoint. This is based on the ability of RPA to signal via ATRIP to the ATR kinase. This work is nicely performed. Only a couple of alleles with minor checkpoint defects were identified. Presumably, none of these point mutations is sufficient to disrupt ATRIP signally; presumably this could be observed with a Rad26-GFP fusion.

The authors should cite, and discuss, work from cerevisiae describing RPA as itself a checkpoint substrate, meaning it may function downstream as well as upstream, potentially confounding the analysis. Aside from these minor concerns the work is adequately done.

Reviewer #3: Xu et al described the identification of fission yeast Rpa1 mutations that impair DNA replication checkpoint (DRC) but not the DNA damage checkpoint (DDC). RPA is involved many DNA transactions, such as DNA replication, DNA repair pathways, and checkpoint responses. Commonly used RPA mutant alleles in budding and fission yeasts tend to have defects in many processes and manifest highly complex phenotype. When RPA mutants impair DNA replication and repair, cells sustain more lesions, which can lead to stronger DDC and DRC responses. Consequently, despite the widely held belief that RPA is needed for DDC/DRC, available RPA mutants do not reduce DDC and DRC levels. The authors confirmed proficient DDC/DRC in ssb1-R229H, D223Y, and G79E mutants (all have ≥ 2X protein level reduction) in HU and MMS, and that they showed chronic but not acute sensitivity toward HU.

Addressing the contradiction requires separation-of-function RPA alleles that specifically impair DDC and/DRC. The authors first tried to generate rfa1-t11 (K45E) equivalent mutations, but three types of mutations all caused lethality or severe growth defect. Thus, they performed PCR-based mutagenesis of Ssb1 and integrated the mutated alleles into chromosome using pop-in and pop-out method. Among 25 mutants, the authors found six showed some defects suggestive for DRC impairment, further analyses using integrated alleles found that ssb1-1 (K33E, Y264H) and -10 (K421R-Y474N-T585I) met the DRC impairment criteria better than to others, and also showed stronger bleomycin sensitivity than rad3∆ and exhibit slower growth, suggesting for replication and DSB repair defects.

While the data suggest that the ssb alleles were not be the ideal separation of function alleles, demonstrating they indeed reduce DRC helps to address the conundrum described above regarding the role of RPA in checkpoint. The study was done thoroughly and both the results and mutants generated can be useful to the field. I have two main suggestions.

1) Given the large numbers of RPA mutants examined, it is useful to map some of them, particularly, ssb1-1 and 10, on a 3D structure of RPA, which can be generated by the alpha-fold program. Are the mutations located in the area involved in binding to ssDNA and/or ATRIP/Rad26/Ddc2? This excise may help to provide plausible explanations regarding why specific alleles impair DRC while many others do not.

2) ssb1-1 and -10 likely affect DNA replication and break repair; authors might want to try to separate the mutations within each allele and see if a true separation of function allele with specific DRC defect can be obtained. The authors seemed to have already generated such alleles, it would be informative to include those data/

Minor point. - Figure 1 needs to be made into two separate figures.

**Have all data underlying the figures and results presented in the manuscript been provided?**

Reviewer #1: Yes

Reviewer #2: Yes

Reviewer #3: Yes

PLOS authors have the option to publish the peer review history of their article (what does this mean?). If published, this will include your full peer review and any attached files.

Reviewer #1: No

Reviewer #2: No

Reviewer #3: No

---

## [Decision Letter · Decision Letter 1]

24 Apr 2023

Dear Dr Xu,

We are pleased to inform you that your manuscript entitled "Comprehensive mutational analysis of the checkpoint signaling functions of Rpa1/Ssb1 in fission yeast" has been editorially accepted for publication in PLOS Genetics. Congratulations!

Yours sincerely,

Geraldine Butler

Section Editor

PLOS Genetics

Comments from the reviewers (if applicable):

Reviewer's Responses to Questions

**Comments to the Authors:**

Reviewer #1: All my concerns are addressed well in the revised version.

Reviewer #3: Inclusion of Figure S11 is useful but would be more insightful to provide some explanation of effects on three other point mutations based on the location and conservation of the residues.

**Have all data underlying the figures and results presented in the manuscript been provided?**

Reviewer #1: Yes

Reviewer #3: Yes

PLOS authors have the option to publish the peer review history of their article (what does this mean?). If published, this will include your full peer review and any attached files.

Reviewer #1: No

Reviewer #3: No

**Data Deposition**

http://datadryad.org/submit?journalID=pgenetics&manu=PGENETICS-D-23-00250R1

**Press Queries**

---

## [Editor Report · Acceptance letter]

15 May 2023

PGENETICS-D-23-00250R1 

Comprehensive mutational analysis of the checkpoint signaling functions of Rpa1/Ssb1 in fission yeast 

Dear Dr Xu, 

We are pleased to inform you that your manuscript entitled "Comprehensive mutational analysis of the checkpoint signaling functions of Rpa1/Ssb1 in fission yeast" has been formally accepted for publication in PLOS Genetics! Your manuscript is now with our production department and you will be notified of the publication date in due course.

With kind regards,

Zsofi Zombor

PLOS Genetics

On behalf of:
